


# Sedimentary alkalinity generation and long-term alkalinity development in the Baltic Sea

Erik Gustafsson[1], Mathilde Hagens[2,3], Xiaole Sun[4], Daniel C. Reed[5], Christoph Humborg[4,6,7], Caroline P. Slomp[2], Bo G. Gustafsson[1,7]

[1] Baltic Nest Institute, Baltic Sea Centre, Stockholm University, SE-10691, Stockholm, Sweden
[2] Department of Earth Sciences, Geochemistry, Utrecht University, P.O. Box 80.021, 3508 TA Utrecht, the Netherlands
[3] Now at: Soil Chemistry and Chemical Soil Quality, Wageningen University, P.O. Box 47, 6700 AA Wageningen, the Netherlands
[4] Baltic Sea Centre, Stockholm University, SE-10691, Stockholm, Sweden
[5] Fisheries & Oceans Canada, Bedford Institute of Oceanography, Canada
[6] Department of Environmental Science and Analytical Chemistry, Stockholm University, SE-10691, Stockholm, Sweden
[7] Tvärminne Zoological Station, University of Helsinki, J.A. Palménin tie 260, 10900 Hanko, Finland

*Correspondence to*: Erik Gustafsson (erik.gustafsson@su.se), Mathilde Hagens (mathilde.hagens@wur.nl)

**Abstract.** Enhanced release of alkalinity from the seafloor, principally driven by anaerobic degradation of organic matter under low-oxygen conditions and associated secondary redox reactions, can increase the carbon dioxide ($CO_2$) buffering capacity of seawater and therefore oceanic $CO_2$ uptake. The Baltic Sea has undergone severe changes in oxygenation state and total alkalinity (TA) over the past decades. The link between these concurrent changes has not yet been investigated in detail. A recent system-wide TA budget constructed for the past 50 years using BALTSEM, a coupled physical-biogeochemical model for the whole Baltic Sea area, revealed an unknown TA source. Here we use BALTSEM in combination with observational data and one-dimensional reactive transport modelling of sedimentary processes in the Fårö Deep, a deep Baltic Sea basin, to test whether sulfate reduction coupled to iron (Fe) sulfide burial can explain the missing TA source in the Baltic Proper. We calculated that this burial can account for 26% of the missing source in this basin, with the remaining TA possibly originating from unknown river inputs or submarine groundwater discharge. We also show that temporal variability in the input of Fe to the sediments since the 1970s drives changes in sulfur burial in the Fårö Deep, suggesting that Fe availability is the ultimate limiting factor for TA generation under anoxic conditions. The implementation of projected climate change and two nutrient load scenarios for the 21st century in BALTSEM shows that reducing nutrient loads will improve deep water oxygen conditions, but at the expense of lower surface water TA concentrations, $CO_2$ buffering capacities and faster acidification. When these changes additionally lead to a decrease in Fe inputs to the sediment of the deep basins, anaerobic TA generation will be reduced even further, thus exacerbating acidification. This work highlights that Fe dynamics play a key role in the release of TA from sediments where Fe sulfide formation is limited by Fe availability, as exemplified for the Baltic Sea. Moreover, it demonstrates that burial of Fe sulfides should be included in TA budgets of low oxygen basins.





# 1 Introduction

Assimilation of $CO_2$ by autotrophs followed by sedimentation and burial of organic carbon is a sink for atmospheric $CO_2$ (Sarmiento and Gruber, 2006). Large proportions of global oceanic primary production, organic matter burial, and sedimentary mineralization occur in coastal seas (Gattuso et al., 1998). Despite covering only ca. 7% of the oceanic surface area, coastal

seas contribute ca. 10 to 20% of the global oceanic $CO_2$ uptake (Gattuso et al., 1998; Bauer et al., 2013; Regnier et al., 2013). One effect of eutrophication, the increased supply of organic matter to an ecosystem, is that $CO_2$ assimilation as well as burial of carbon (C) is enhanced (Andersson et al., 2006; Middelburg and Levin, 2009). In addition, eutrophication drives an accelerated deep water deoxygenation in many coastal systems (Diaz and Rosenberg, 2008; Rabalais et al., 2015). Because increased mineralization of organic matter leads to enhanced $CO_2$ release, eutrophication-induced hypoxia may intensify

acidification in sub-surface waters of such coastal systems (Cai et al., 2011, 2017; Hagens et al., 2015; Laurent et al., 2018). Enhanced deep water oxygen consumption may also increase the proportion of organic matter that is degraded anaerobically in both sediments and deep water. Many anaerobic degradation processes produce total alkalinity (TA) (Chen and Wang, 1999), which can temporarily or permanently boost the pelagic $CO_2$ buffering capacity and thus potentially increase the absorption of atmospheric $CO_2$ (or reduce $CO_2$ outgassing). Estimates of TA release from coastal sediments have been based

both on model calculations and direct measurements, but the reported fluxes vary quite considerably depending on the methods used, processes included, and spatial and temporal scales considered (Chen, 2002; Wallmann et al., 2008; Thomas et al., 2009; Hu and Cai, 2011a; Krumins et al., 2013; Gustafsson et al., 2014b; Brenner et al., 2016).

Depending on the nitrogen (N) source, primary production can either be a source, a sink, or neutral with respect to TA (Wolf-Gladrow et al., 2007). Aerobic mineralization including nitrification of the produced ammonium is a TA sink, while anaerobic

mineralization processes in general produce TA (e.g. Brenner et al., 2016). On a system scale, however, the ultimate buildup of TA due to primary production and mineralization depends on the source of the reactants and/or the fate of the products of all alkalinity-generating/consuming reactions. For example, production of dinitrogen gas ($N_2$) during pelagic or benthic denitrification results in a permanent loss of nitrate and hence a gain of TA (Soetaert et al., 2007). This process only results in net TA production on a system scale, however, if the nitrate is derived from an external source rather than from local

nitrification (Hu and Cai, 2011b). Similarly, sulfate reduction leads to net TA generation only if the produced sulfide is buried as e.g. iron (Fe) sulfides rather than being reoxidized within the same system (Hu and Cai, 2011a). Note that the location of sulfide reoxidation, i.e. sediment or water column, impacts net TA generation in the sediments but not on a system scale.

The Baltic Sea (Fig. 1) is one of many coastal seas around the globe where eutrophication has led to massive changes in both nutrient cycling and oxygen concentrations (e.g. Gustafsson et al., 2012). During the first half of the 20th century, hypoxic and

anoxic conditions occurred only sporadically and affected limited deep water areas (Carstensen et al., 2014). Since the 1950s, oxygen poor areas in the Baltic Sea have expanded rapidly and today form the largest anthropogenic 'dead zone' in the world (Diaz and Rosenberg, 2008). This expansion may have led to an increase in net TA generation through anaerobic processes.




Present-day riverine TA loads to the Baltic Sea amount to ~470 Gmol y$^{-1}$ according to available observations (Gustafsson et al., 2014b). Based on budget calculations, Gustafsson et al. (2014b) estimated that an additional TA source of 344 Gmol y$^{-1}$ is necessary to close the Baltic Sea TA budget. Approximately 260 Gmol y$^{-1}$ of this source cannot be explained so far. Of the 84 Gmol y$^{-1}$ that was resolved, 66 Gmol y$^{-1}$ resulted from the net effect of primary production, aerobic mineralization, and

denitrification – essentially N cycling. The remaining 18 Gmol y$^{-1}$ resulted from net sulfate reduction (sulfate reduction – sulfide oxidation) in the water column, but this fraction could be reversed in case of oxygenation of the water column. It was hypothesized that a significant fraction of the unresolved TA source could be coupled to burial of Fe sulfides as a result of anaerobic mineralization in sediments. Due to incomplete descriptions of benthic processes in the model that was used, this hypothesis could not be tested (Gustafsson et al., 2014b), but the process has recently been identified as an important TA

source in the Gdańsk Deep (Lukawska-Matuszewska and Graca, 2018).

The amount and form of Fe solids entering the sediment is a key factor controlling net benthic TA generation from Fe sulfide burial. Recent work on Fe dynamics in deep Baltic Sea basins has shown that the lateral transfer ("shuttling") of Fe from shelves to deep basins is most intense when bottom water hypoxia is intermittent (Lenz et al, 2015a). Under such conditions, dissolved Fe can escape from the shelves, rather than being retained in the sediment as Fe oxides (in case of oxic bottom water

conditions) or Fe sulfides (in case of widespread anoxia/euxinia). This Fe is then transported laterally to the deep basins, where local redox conditions determine its fate. The present-day low oxygen concentrations in many deep basins of the Baltic Sea promote sulfate reduction (Reed et al., 2016), indicating enhanced net benthic TA generation due to sulfur (S) burial as the escaped Fe reaches these basins. Sediment records of S concentrations can be used to calculate S burial and thus quantify the TA source associated with this burial.

Here, we use burial rates of solid phase S from the literature (Lenz et al. 2015b) and results from two different types of biogeochemical models to 1. Constrain the present-day sedimentary TA release from Baltic Sea sediments to the water column; 2. Quantify the large-scale changes in sedimentary TA release coupled to changes in eutrophication and oxygen conditions; 3. Quantify the relative influence of different processes that contribute to the sedimentary TA release and 4. Estimate the potential future development of TA and pH levels upon recovery from eutrophication and assuming continued eutrophication. The

models employed in this study are a high-resolution reactive-transport sediment model (RTM) (Reed et al. 2016) and a long-term, large-scale coupled physical-biogeochemical model for the Baltic Sea, BALTSEM (Gustafsson et al., 2017).

## 2 Material and methods

### 2.1 Data

#### 2.1.1 Sediment data and calculations

Sedimentary alkalinity generation due to S burial in the Baltic Proper was estimated using published data of S contents at F80, a 191 m deep site in the Fårö Deep of the Gotland Sea (Fig. 1; Lenz et al., 2015b). Concentrations of S in µmol g$^{-1}$ were



converted to units of µmol cm$^{-3}$ using measured porosities and were depth-integrated to 25 cm. Following the age model presented by Lenz et al. (2015b), this depth interval represents the burial since 1970, allowing the calculation of an annually averaged rate of S burial (mmol m$^{-2}$ y$^{-1}$). Using a 1:2 ratio between S burial and TA generation, the latter was calculated and subsequently extrapolated to the basin scale using the total muddy sediment area for the Baltic Proper (Table 1; Al-Hamdani and Reker, 2007).

Although total S concentrations do not indicate in which form S is buried, this does not matter for the associated TA generation. The conversion from sulfate ($SO_4^{2-}$) to reduced sulfur produces 2 moles of TA (in the form of $HCO_3^-$) per mole of S (Eq. 1, $CH_2O$ represents simplified organic matter). This is irrespective of whether it ultimately ends up in the form of Fe monosulfides (FeS), pyrite ($FeS_2$) or elemental sulfur ($S^0$) when being converted to a solid form. Reductive dissolution of Fe oxides also produces 2 moles of TA (as $HCO_3^-$) per mole of dissolved iron ($Fe^{2+}$) formed, but this is compensated for when $Fe^{2+}$ subsequently reacts with dihydrogen sulfide ($H_2S$) during FeS formation, thereby releasing protons (Eq. 2). Therefore, there is no net TA generation associated with the formation of $Fe^{2+}$ and its subsequent burial as Fe sulfide minerals (Hu and Cai, 2011a).

$$2CH_2O + SO_4^= \rightarrow 2HCO_3^- + H_2S \tag{1}$$

$$\tfrac{1}{4}CH_2O + Fe(OH)_3 + H_2S + \tfrac{7}{4}CO_2 \rightarrow FeS + \tfrac{3}{4}H_2O + 2HCO_3^- + 2H^+ \tag{2}$$

### 2.1.2 Oceanographic data

Measured TA and salinity in the 1970-2014 period were extracted from the ICES oceanographic database (ICES Data Set on Ocean Hydrography, the International Council for the Exploration of the Sea, Copenhagen, http://ocean.ices.dk/Helcom/) and the Swedish Ocean Archive (SHARK) database provided by the Swedish Meteorological and Hydrological Institute (SMHI; http://sharkweb.smhi.se/).

During the Swedish monitoring cruises water samples were at least occasionally stored in glass bottles with a head space of air until later analysis in the laboratory. This means that the dissolved sulfide ($\Sigma H_2S$) in the samples may have been oxidized by the time of analysis (cf. Ulfsbo et al., 2011), which then implies that the reported TA concentrations in anoxic water can be substantially underestimated. Following Ulfsbo et al. (2011) we have for that reason adjusted the measured TA concentrations in euxinic waters by adding the $\Sigma H_2S$ concentration multiplied by a factor two (i.e., $TA_{adjusted} = TA_{observed} + 2\ \Sigma H_2S_{observed}$). However, if the hydrogen sulfide was not removed by the time of analysis, the adjusted TA concentration will be too high.

### 2.1.3 River data

Riverine TA concentrations in the BALTSEM model were based on monthly measurements in 1996-2000 from 82 of the major rivers entering the Baltic Sea, representing approximately 85% of the total runoff (cf. Gustafsson et al., 2014b). In this study we also include measurements from Swedish and Finnish rivers for the periods 1985-2012 and 2001-2012, respectively. Swedish chemical data were provided by the Swedish University of Agricultural Sciences (SLU; http://www.slu.se/en/),



Swedish runoff data were provided by the SMHI (http://vattenwebb.smhi.se/). Finnish data were extracted from the database Hertta provided by the Finnish Environment Institute (SYKE).

## 2.2 Model calculations

### 2.2.1 Sediment reactive-transport model (RTM)

A one-dimensional reactive-transport model (Reed et al., 2016) was used to calculate benthic TA generation and release at F80, with a minor modification: the redox reaction equations as presented in Table S8 by Reed et al. (2016) were updated to include the total concentrations of the ammonium and sulfide acid-base systems, instead of the respective corresponding acid-base species. The model calculates acid-base speciation using the Direct Substitution Approach (Hofmann et al., 2008), where pH and total quantities (i.e., dissolved inorganic carbon (DIC), $\Sigma H_2S$, total ammonium ($\Sigma NH_4^+$), etc.) are used as state variables,

meaning that TA is calculated as output variable. Effluxes of TA from the sediment were subsequently calculated from the gradient in the diffusive boundary layer (Boudreau, 1997). The model includes the carbonate, sulfide, ammonium, and phosphate acid-base systems, such that TA is defined as:

$$TA = [HCO_3^-] + 2[CO_3^=] + [HPO_4^=] + 2[PO_4^{3-}] + [NH_3] + [HS^-] - [H^+] \qquad (3)$$

The RTM thus ignores contributions of the fluoride, borate and silicate acid-base systems, as well as the hydroxide ion and

phosphoric acid, which make up part of the classical definition of TA (Dickson, 1981) but were expected to have a low contribution to TA in this setting. Moreover, the model neglects organic alkalinity, which can substantially contribute to Baltic Sea pore water TA (Lukawska-Matuszewska, 2016; Lukawska-Matuszewska et al., 2018), but which is challenging to calculate due to the variety of acid-base groups associated with organic matter. Further details on the governing equations, redox and equilibrium reactions, reaction parameters and boundary conditions are given in Eq. S1-S22 and Table S1 (supplementary

material), and in Reed et al. (2016).

The RTM has previously been used at this location to assess the impact of shelf-to-basin iron shuttling on the formation and stability of the Fe(II)-phosphate mineral vivianite (Fe$_3$(PO$_4$)$_2$·8H$_2$O; Reed et al., 2016). To this end, it was calibrated against a wide selection of pore water and solid phase data presented in Jilbert and Slomp (2013), Lenz et al. (2015b) and Reed et al. (2016). We confirm this calibration for the carbonate system with additional DIC and TA pore water data from a multicore

recovered at F80 during a research cruise with R/V Pelagia in June 2016. Core handling and pore water analyses have been performed following Egger et al. (2016). We used the previous calibration and perform sensitivity analyses to identify the key mechanisms responsible for benthic TA generation and release. To represent the variety of bottom water conditions at site F80 since the 1970s, four time intervals were recognized (Fig. 2): 1970-1973 (baseline; I), 1973-1978 (start of change in Fe loading; II), 1978-1981 (eutrophication but pre-euxinia; III) and 1981-2009 (eutrophication and euxinia, IV).



### 2.2.2 Large scale physical-biogeochemical model

BALTSEM is a coupled physical-biogeochemical model developed for the Baltic Sea. The model divides the system into thirteen connected sub-basins (Fig. 1), where each basin is described as horizontally homogeneous although with a high vertical resolution and a depth dependent area distribution based on the real hypsography of the various sub-basins. A hydrodynamic

module simulates mixing and advection (Gustafsson, 2000; 2003), while the dynamics of nutrients and plankton (Gustafsson et al., 2012; Savchuk et al., 2012; Gustafsson et al., 2017) as well as organic carbon and the carbonate system processes (Gustafsson et al., 2014a, b; 2015) are simulated in a coupled biogeochemical module. The BALTSEM model runs cover the period 1970-2014 in the present study.

Most biogeochemical processes related to production and mineralization in the water column and sediments either produce or

consume TA. Many of these TA sources and sinks have been described in detail by e.g. Wolf-Gladrow et al. (2007) and Krumins et al. (2013) respectively. TA production/consumption resulting from processes such as ammonium/nitrate-based production, nitrification, denitrification, sulfate reduction, and sulfide oxidation are included in the BALTSEM calculations (Gustafsson et al., 2014b). Under euxinic conditions in the water column, sulfate reduction and also ammonium accumulation represent large TA sources, these are however reversed if the water is again oxygenated by deep water inflows and vertical

mixing. BALTSEM does not include Fe cycling, and in particular there are no parameterizations for Fe shuttling and subsequent burial of Fe sulfides in the sediments.

Other TA sources than river loads are, as mentioned in the introduction, necessary to reproduce observed TA in the Baltic Sea. The additional sources can partly be explained by biogeochemical processes but are as yet largely unknown. These "unresolved sources" were calibrated by Gustafsson et al. (2014b), and although the magnitude of the unresolved source is well constrained,

it has as yet not been possible to determine what this source is. In theory the source could be associated both with processes that are not included in the model (e.g. Fe-S cycling, submarine groundwater discharge, etc.) and with processes that are included but possibly not correct (e.g. river loads, nutrient cycling). The BALTSEM model has since been updated both with new processes and with new forcing files. The model now includes the influence from organic alkalinity in the water column based on Kuliński et al. (2014) and Ulfsbo et al. (2015) (cf. Gustafsson et al., 2015), as well as the influence from acidic

depositions based on Claremar et al. (2013). Furthermore, the forcing files now cover the period 1970-2014. As a result of these updates, the calibrated unresolved TA sources have been slightly modified in the present study compared to those by Gustafsson et al. (2014b) (see Section 3.2).

Although the processes behind the unresolved TA source are not known, the calibrated sources in the different sub-basins are tentatively added as a flux from the sediment to the water column. Following Gustafsson et al. (2014b), no additional

unresolved TA sources were added to the Kattegat and Danish Straits (sub-basins 1-6, Fig. 1). Since these basins have quite short residence times (Gustafsson, 2000), internal TA generation will not significantly influence concentrations from conservative mixing between inflowing saline water from the North Sea and outflowing fresher waters from the Baltic Proper.



For that reason, it is not feasible to constrain any unresolved sources in these areas, although the same processes that generate TA in the remaining Baltic Sea should apply to these sub-basins as well.

### 2.2.3 Merits and limitations of using two models

Using a mass-balance approach, BALTSEM connects external sources, transports between basins, and internal cycling of carbon, nutrients, and TA within each sub-basin. The model is thus highly useful to quantify fluxes – resolved as well as unresolved – on a basin- and system-scale. It is furthermore an invaluable tool when investigating multi-stressor effects on the ecosystem in future scenario calculations. However, the lack of parameterizations for TA production and consumption related to sedimentary Fe-S cycling means for example that there is no S burial – a process that represents a net TA source. The RTM on the other hand resolves these processes in detail and quantifies the fluxes at specific sites. At present, it is not feasible to upscale such site-specific fluxes to the system-scale – in the present study fluxes computed by the RTM are instead upscaled to cover a certain bottom type in the relevant sub-basin (cf. Section 2.1.1).

This approach means that it is for the first time possible to estimate how benthic Fe-S processes influence TA in the Baltic Sea, and in particular to what extent the previously mentioned unresolved sources can be associated with S burial. Other processes that influence TA (e.g. redox reactions involving N) are included in both models. Although benthic N cycling is described in more detail in the RTM, it is in this case preferable to use the fluxes as calculated by BALTSEM. One reason is to take advantage of the coupled physical-biogeochemical approach as described above, so that the source of the nitrate can be identified on a basin scale. On top of that, denitrification rates at F80 are not representative for the entire sub-basin.

Ideally the RTM would be dynamically coupled to BALTSEM, but this is currently not feasible for two reasons: First, BALTSEM has approximately 1400 sediment "boxes", and the RTM would have to compute the sediment processes in each of these boxes – resulting in a great computational cost. Second, calibration of the RTM in various parts of the Baltic Sea would be problematic because of an insufficient coverage of sediment data. Thus, the two models are not coupled to one another but instead used independently.

## 3 Results

### 3.1 Sediment and RTM calculations

Both the model- and observation-based estimates indicate that between 1970 and 2009, on average 291-295 mmol S m$^{-2}$ has annually been buried, leading to a TA generation of 582-590 mmol m$^{-2}$ y$^{-1}$. This corresponds to a TA flux of ~43.2-43.8 Gmol y$^{-1}$ from muddy sediments (Table 2). The model further suggests that virtually all of the S solids are present in the form of FeS$_2$. Of the 291 mmol m$^{-2}$ y$^{-1}$ of S being buried, only 56% was formed in situ, whereas the remaining 44% was deposited as a result of the shuttling of Fe in the form of FeS$_2$ to the deep basin (Lenz et al., 2015a). However, as BALTSEM does not resolve FeS$_2$ formation in either the water column or sediment, both need to be included when estimating the unresolved TA source due to sulfate reduction and S burial.



In line with other studies (e.g. Jørgensen, 1977), the vast majority of reduced S produced through sulfate reduction was reoxidized in either the sediment or the overlying water. On average, only 10.2% was buried, but there was strong temporal variability in this percentage (Table 3). The fraction of S solids being buried was highest under eutrophic but non-euxinic conditions (i.e. period III; 40.7%). Since 1981 (period IV), inputs of Fe oxides have decreased, leading to a higher efflux of

$\sum H_2S$ and thus less S burial, despite higher sulfate reduction rates (SRR). Our results indicate that even under non-euxinic conditions, pyrite formation was limited by the availability of highly reactive iron, as is the case in most marine systems (Berner, 1984; Raiswell and Canfield, 2012) This limitation was confirmed by the difference between potential and simulated S formation rates (Table 3), where the former indicates the amount of solid S that could have formed based on the other modeled sources and sinks of $\sum H_2S$. It is thus indicative of the amount of S mineral formation under unlimited $Fe^{2+}$ supply.

In the period 1970-2009, S burial could on average only explain 328 mmol $m^{-2}$ $y^{-1}$ of TA generation (Table 3; using a 1:2 ratio between S burial and net TA generation). This was 9.2% of the internally generated TA (3948 mmol $m^{-2}$ $y^{-1}$), but again clear temporal variations were observed (Table 4). Under the baseline conditions (period I), when little S was buried, it only made up 3.7% of the total TA generation. This percentage increased to 12.4% between 1973-1978 (period II), when more Fe was available, and peaked at 34.5% between 1978-1981 (period III), when both $Fe(OH)_3$ and carbon loadings were high. Since

1981 (period IV), the decrease in $Fe(OH)_3$ loading has further limited S burial, leading to a contribution of only 6.7% of the internally generated TA.

Pore-water profiles of DIC and TA (Fig. 3) indicate that the model is well calibrated for the carbonate system. The 2016 DIC data show a better fit with the modeled profile than the previously published DIC data (Reed et al., 2016), while the reverse is true for both TA data sets. The profiles furthermore show that the model overestimates the buildup of $\sum H_2S$. This can be

explained by loss of $\sum H_2S$ during sampling, which is a common problem for anoxic sediments, but also by the chosen lower boundary condition of the model. Whereas the model assumes no gradient with the underlying sediment, the data (Jilbert and Slomp, 2013; Lenz et al., 2015b) show a declining trend with depth below 32 cm, suggesting a downward diffusive flux of ~144 mmol $\sum H_2S$ $m^{-2}$ $y^{-1}$. Rate profiles of the most important processes contributing to TA (Fig. 3) however show that this buildup of $\sum H_2S$ did not substantially affect sedimentary TA generation, as only marginal amounts of $\sum H_2S$ reacted below 10

cm. These rate profiles furthermore show that especially between 1978-1981 (period 3), when OM and Fe inputs were high but bottom waters were still oxic, intense cycling of Fe occurred in the sediments, associated with high TA production and consumption. Dissolved Fe produced from reductive dissolution of amorphous iron oxides during OM degradation either diffused upward, where it reoxidized in the oxic sediments, or downward to react with $\sum H_2S$. Well-crystalline iron oxides, assumed to be inaccessible for OM degradation, reacted with $\sum H_2S$ over a wide range, thereby producing additional $Fe^{2+}$.

On a system scale, this cycling of Fe does not lead to net TA generation (Hu and Cai, 2011a), but it may impact the efflux of alkalinity from the sediment that is calculated by the RTM. This flux cannot directly be used to assess the long-term net TA generation that we are interested in, as it is the product of a variety of reversible and irreversible TA generating reactions, such as the intense Fe cycling discussed above. Moreover, its constituents (e.g. $HS^-$) may become reoxidized in the water column. However, the magnitude and temporal variability of the efflux compared to those of S burial and total TA generation may




provide information on its driving processes. Note that the difference between total TA generation and efflux (Table 4; Fig. 4) reflects the buildup of TA in the sediment, as well as loss of TA at depth through burial.

A comparison of their temporal variabilities shows that the benthic TA efflux only partly followed the pattern in S burial (Fig. 4; Table 3). Since 1973, when the efflux was 1901 mmol m$^{-2}$ y$^{-1}$, it decreased to 1561 mmol m$^{-2}$ y$^{-1}$ in 1978, followed by a
sharp increase to 3261 mmol m$^{-2}$ y$^{-1}$ in 1982 and a more gradual increase to 4823 mmol m$^{-2}$ y$^{-1}$ in 2009. Generation of TA throughout the entire sediment column contributed to the calculated efflux (Table 4), to a major extent due to high rates of sulfate reduction at depth (Fig. 3). The methane diffusing upward from deeper sediment layers played a key role here, being responsible for on average ~95% of the CH$_4$-driven sulfate reduction and ~43% of the total sulfate reduction. The temporal change in spatial pattern of the CH$_4$-driven sulfate reduction (Fig. 3) suggests that the role of upward diffusing CH$_4$ relative to
in situ produced CH$_4$ has become less important over time, as in 2009 highest reaction rates were found at ~8 cm depth rather than at the bottom of the modeled sediment column.

Interestingly, the decrease in efflux between 1973-1978 (period II), which resulted from changes in the iron loading, was not mimicked in either the total TA generation or the amount of S burial, but rather reflected the pattern of the change in TA generation through secondary reactions (Fig. 4). The most important secondary reaction contributing negatively to TA between
1973-1978 was the reoxidation of Fe$^{2+}$, the rate of which more than doubled during this period (Table 4; Fig. 3) and which, in contrast to the other dominant reactions, was restricted to the upper cm of the sediment column (Fig. 3). This indicates that it was the driving force of the lower TA efflux during this period. Although reoxidation of Fe$^{2+}$ consumed even more TA in period III (1978-1981), this was more than compensated for by the concurrent enhanced TA generation due to OM degradation, especially coupled to iron oxide reduction, even though that occurred deeper in the sediment (Fig. 3; Table 4).

**3.2 BALTSEM calculations**

The recalibrated unresolved TA sources as well as the resolved pelagic and benthic TA sources minus sinks in the different sub-basins as calculated with BALTSEM are indicated in Table 5. In total, the unresolved source amounts to 257.5 Gmol y$^{-1}$, while the total resolved pelagic and benthic sources minus sinks amount to a net source of 119.4 Gmol y$^{-1}$ over the 1970-2014 period. For comparison, the riverine TA load amounts to 469.5 Gmol y$^{-1}$. In Fig. 5-6, simulated and observed surface and deep
water TA normalized to mean salinity at the corresponding station and water depth (TA$_N$) and salinity are shown. The normalized TA is used in order to avoid uncertainties related to discrepancies between simulated and observed salinity.

The temporal development of resolved and unresolved TA sources minus sinks throughout the model simulation are shown in Fig. 7. In this simulation, the unresolved sources in the different sub-basins were assumed to remain constant throughout the model run (Fig. 7), while the resolved sources and sinks vary depending on primary productivity, oxygen conditions, denitrification rates, and other biogeochemical processes included in the model (Fig. 5-6). Despite the constant unresolved
sources, simulated TA$_N$ concentrations generally reproduce observed values. Exceptions are the simulated TA$_N$ concentrations in the Kattegat and the Gotland Sea in the 1980's, where actual concentrations are overestimated, and TA$_N$ concentrations in the Bothnian Sea and Bay that are underestimated in the last ten-year period (Fig. 5-6).



There is an overall long-term increase in the resolved net TA generation in sediments and water column combined (Fig. 7), reflecting the ongoing eutrophication and overall deteriorating oxygen conditions of the Baltic Sea. The resolved net pelagic TA source increases in the period 1970-2000 in response to an increased primary production and then levels out and slightly declines in the last decade. The increased resolved benthic source in the last decade on the other hand is a response to

deteriorating oxygen conditions resulting in increased TA generation through denitrification and sulfate reduction. Sulfate reduction in the BALTSEM model is however not an irreversible source, since sulfidic waters can be reoxidized by deep water inflows, thus consuming TA and reversing the source.

## 4 Discussion

### 4.1 Use of BALTSEM and RTM in the context of this work

Given the detailed presentation of sedimentary processes and effluxes in section 3.1, one may wonder why only S burial is used in the coupling to BALTSEM. After all, the RTM calculations include many processes other than S burial. However, to study the impact of sedimentary TA generation on the long-term TA development in the Baltic Sea, we need to take into account only those processes that are relevant to accomplish this task.

BALTSEM includes many processes, producing and consuming TA both reversibly and irreversibly on short time scales and

in many boxes within each sub-basin of the Baltic Sea. The net result of these simulations, i.e. the long-term development of TA in various sub-basins, is what we compare to observations. Similarly, the RTM calculates net TA generation due to various reversible and irreversible processes. If we dynamically coupled the RTM to BALTSEM, we would have to consider all these processes, and link all species between both models. Given the unfeasibility of this, as discussed in section 2.2.3, we couple both models by using the output of the RTM to further constrain BALTSEM. Specifically, we explain part of the source of

BALTSEM that is unresolved but necessary to describe the long-term TA development in the Baltic Sea. This means that in this context we only need to consider the processes from the RTM that are a) irreversible on the time scale of interest (i.e., decades); and b) not included in BALTSEM. Burial of Fe sulfides (Hu and Cai, 2011a) is the only major process that falls in this category. Denitrification using an external $NO_3^-$ source, the other main pathway for net TA generation (Hu and Cai, 2011b) is already included in BALTSEM. Many other sedimentary processes produce or consume TA (Eq. S1-S22, supplementary

material; Soetaert et al., 2007), but they are not irreversible on the relevant time scale. Their dynamics are, however, highly interesting to discuss as they help determine what limits net sedimentary TA generation, and which processes mainly drive the effluxes of TA and other constituents to the water column. Note that this irreversibility is also a reason why we do not use these effluxes as input to BALTSEM. In addition, they are already partly included in BALTSEM, e.g. in the case of $\sum H_2S$ produced from sulfate reduction.





## 4.2 Sulfur burial and TA generation in the Baltic Proper

While mineralization in the sediments occurs everywhere where there is labile organic matter, permanent burial of organic matter as well as other solids such as Fe sulfides should predominantly occur in muddy sediments. Consequently, the part of the unresolved TA source that is a result of sulfur burial should be released from muddy sediments rather than from the entire sediment surface area of the Baltic Sea. Our RTM calculations in combination with observations from site F80 in the Baltic Proper provide the first estimate of TA generation resulting from S burial in Baltic Sea sediments (582-590 mmol m$^{-2}$ y$^{-1}$). Assuming that the calculated TA generation resulting from S burial is representative only for the muddy sediment area in the Baltic Proper (74300 km$^2$; Table 1), the total annual flux in this area is ~44 Gmol y$^{-1}$. The calibrated unresolved TA source in the Baltic Proper amounts to 166 Gmol y$^{-1}$ according to the BALTSEM model (Table 5). Based on the RTM calculations, TA generation coupled to sulfur burial could thus cover some 26% of the unresolved source at least in this sub-area of the system. The remaining unresolved TA source of ~74% could possibly be explained by underestimated river loads or submarine groundwater discharge of TA (e.g. Szymczycha et al., 2014). We have no data to quantify these fluxes, however.

In the BALTSEM simulation, the unresolved TA source in the Baltic Proper corresponds to a flux of 730 mmol m$^{-2}$ y$^{-1}$ if the source is distributed evenly over the entire sediment surface (228000 km$^2$). However, if the unresolved source is instead constrained only to muddy sediments, the flux would amount to 2236 mmol m$^{-2}$ y$^{-1}$, which is far above the long term mean flux due to S burial as obtained by RTM calculations. Even during peak pyrite formation periods, S burial only resulted in a source of 1078 mmol TA m$^{-2}$ y$^{-1}$ (Table 3). Furthermore, in a BALTSEM experiment where the unresolved sources were released only from muddy sediments (but at higher rates corresponding to the smaller surface areas), the deep water TA concentrations in particular in the Baltic Proper were overestimated while the surface water TA concentrations were underestimated (not shown).

## 4.3 Reversible versus irreversible sedimentary processes generating TA

As demonstrated by both simulated and observed sediment profiles at F80, a transition from hypoxic to euxinic conditions around 1980 resulted in a strong increase in both solid phase S and Fe burial (Reed et al., 2016). Furthermore, the molar S to Fe ratio of ~2 suggests formation and burial of mostly FeS$_2$. Both FeS$_2$ and FeS can be formed from reactive Fe$^{2+}$ and sulfide produced during Fe(OH)$_3$ and sulfate reduction, respectively. These redox reactions ultimately result in net TA generation (Eq. S1-S22, supplementary material). Another possible pathway is that methane (formed by methanogenesis) is oxidized anaerobically by reduction of either sulfate or Fe(OH)$_3$ (Slomp et al., 2013; Egger et al., 2015b), and Fe and sulfide can then be sequestered in the form of e.g. FeS$_2$. Results from the RTM indicate that CH$_4$ and organic matter are both important electron donors at F80 CH$_4$ oxidation contributes to on average 43.8% of total SRR, and occurs at greater depth than sulfate reduction through organic matter degradation (Fig. 3). Iron-mediated anaerobic oxidation of methane is not included in the set of reactions of this RTM. Previous work has indicated that this process mainly occurs in organic-poor sediments depleted in SO$_4$$^{2-}$




(Riedinger et al., 2014; Egger et al., 2017). These conditions are not met at F80, rendering an important role for this process unlikely.

Apart from such eutrophication-induced changes in the coupled Fe-S cycling, increasingly euxinic conditions also influence manganese (Mn) sequestration in sediments. Dissolved $Mn^{2+}$ can be sequestered in the form of Mn carbonates ($MnCO_3$). If this occurs, the TA generation associated with the reduction of manganese dioxide ($MnO_2$; 2 moles of TA per mole of $MnO_2$; Eq. S1-S22, supplementary material) is completely compensated by the TA sink associated with carbonate removal. However, under euxinic conditions, manganese sulfide (MnS) can be formed if the sulfide availability exceeds the $Fe^{2+}$ availability (Lenz et al., 2015b). Indeed, long-term sediment records indicate a relation between euxinic periods in Baltic Sea deep waters and burial of Mn sulfides in the forms of both rambergite and alabandite (Lepland and Stevens, 1998). As opposed to burial of $MnCO_3$, burial of MnS results in a net TA generation comparable to that of $FeS_2$ burial. In the RTM, we did not investigate the possible impact of MnS formation on TA generation as the sediment record at F80 does not show substantial Mn enrichments in the surface, despite higher sulfide than $Fe^{2+}$ availability (Lenz et al., 2015b).

Ammonium, sulfide, $Fe^{2+}$ and $Mn^{2+}$ are rapidly oxidized if oxygen is supplied to anoxic waters. The result is a TA sink that compensates the TA generation by anaerobic mineralization. Precipitates such as FeS and $FeS_2$ can also be oxidized but this is generally a slower process, especially in the case of $FeS_2$ (Millero et al., 1987; Wang and Van Cappellen, 1996). Moreover, these S minerals are embedded in organic-rich, $\Sigma H_2S$ producing sediments. This reduces the impact of possible reoxygenation of the sediment for extended periods of time, implying that the TA source that results from S burial is stable. Sediment cores indicate the presence of Fe sulfides – in particular $FeS_2$ – in the top three meters of Gotland Sea deep water sediments (Boesen and Postma, 1988) as well as in the top 10 m of deep Bornholm Basin sediments and the top 27 m of Landsort Deep sediments (Egger et al., 2017), all corresponding to roughly 8000 years. In our model results, $FeS_2$ is the dominant form of S in the sediment. Our work also shows that re-oxidation of reduced S never exceeds 0.5% between 1970-2009, irrespective of whether S solids or the total reduced S pool (i.e., including $\sum H_2S$) is investigated (Table S2, supplementary material).

Vivianite formation is another process that generates TA in a net sense. The presence of vivianite in sediment cores (Egger et al., 2015a) indicates that this mineral can be stable upon burial. However, vivianite dissolves in the presence of sulfide (Dijkstra et al., 2018), excluding its burial to be a long-term TA source at F80. It could however be a stable TA source at locations where $\sum H_2S$ rather than Fe availability limits pyrite formation, such as the Bothnian Sea (Egger et al., 2015a).

Calcifying organisms that build calcium carbonate ($CaCO_3$) shells have a large influence on the carbonate system in many marine areas, as illustrated e.g. by high TA fluxes related to $CaCO_3$ formation and dissolution in the North Sea (Brenner et al., 2016). $CaCO_3$ formation results in a TA drawdown in the productive layer, and a TA source where the shells are dissolved. Burial of $CaCO_3$ shells is a net TA sink on a system scale. In the Baltic Sea, however, calcifying plankton species are largely absent – likely because of low saturation values of calcite and aragonite in winter (Tyrrell et al., 2008). In the RTM, $CaCO_3$ dissolution and precipitation is included, based on observed sedimentary $CaCO_3$ contents of ~200 µmol $g^{-1}$ (Fig. 3; see Reed et al., 2016 for further details). The prescribed input of 86 mmol $m^{-2}$ $y^{-1}$ in combination with prevailing conditions in the



sediment led to a net TA loss due to CaCO$_3$ dissolution, which is on average only -0.01 mmol m$^{-2}$ y$^{-1}$ (data not shown). For this reason, we have not included the effects of CaCO$_3$ precipitation and dissolution in our analysis.

## 4.4 Long-term development of TA in the Baltic Sea

Several studies indicate essentially linear TA-salinity relations in Baltic Sea surface water (Ohlson and Anderson, 1990;
Thomas and Schneider, 1999; Perttilä et al., 2006; Beldowski et al., 2010). Long-term TA increases that are not connected to salinity are however apparent from observed TA-salinity relations (Fig. 8; Table S4-S5, supplementary material). Furthermore, Müller et al. (2016) computed a 3.4 µmol kg$^{-1}$ y$^{-1}$ surface water TA increase in the Baltic Proper and a 7 µmol kg$^{-1}$ y$^{-1}$ increase in the Gulf of Bothnia over the past two decades.

The resolved pelagic and benthic TA sources minus sinks in the BALTSEM calculations (Fig. 7) on average increase by
approximately 3 Gmol y$^{-1}$ in the 1970-2014 period. Furthermore, the RTM calculations (Table 3) indicate that sulfur burial can increase by a factor of four after a transition from oxic to anoxic/euxinic conditions, and even by an order of magnitude during this transition if both iron oxide and organic matter loadings are enhanced. Thus, the fraction of the unresolved source that is a result of sulfur burial should be quite variable depending on mineralization rates and oxygen conditions in different areas of the Baltic Sea as well as during different periods in time.

Anaerobic mineralization occurs in sediments even if the overlying water is oxic, and for that reason TA release coupled to sulfur burial does not exclusively occur from sediments covered by sulfidic waters. In fact, our results indicate highest S formation rates under eutrophic, but non-euxinic conditions (Table 3). However, large-scale and long-term changes in TA generation related to changes in sulfur burial are mainly expected to occur in areas experiencing transitions between oxic and anoxic conditions and in addition as a result of changes in iron loadings (Lenz et al, 2015a). Hypoxic and anoxic conditions in
the Baltic Sea water column – as well as rapid transitions between oxic and anoxic conditions – occur primarily in the deep basins of the Baltic Proper, although episodes of oxygen depletion can also occur in the deep water of in particular the Gulf of Finland, as well as in many eutrophic coastal fjords and bays.

The long-term TA decrease in the Gotland Sea in the 1980's coincided with a decreasing salinity (Fig. 5 and S1) as well as improved oxygen conditions in large volumes of the deep water (not shown). During this period, stratification was considerably
weakened and as a result the halocline depth in the Baltic Proper increased, and inflowing new deep water ventilated primarily the upper deep water. Thus, a much larger water volume than usual was well ventilated (e.g. Stigebrandt and Gustafsson, 2007). It is possible that during this period, sulfur burial and associated TA generation was considerably weakened. A very strong TA increase observed in the early 1990's coincides (more or less) with a strengthened stratification due to salt water inflows in 1993. A rapid deterioration of oxygen conditions followed because of a suppressed deep water ventilation during
periods of strong stratification. This development towards increasingly anoxic/euxinic conditions could potentially cause a large response in terms of TA generation.

It is however likely that the observed TA decline in the 1980's followed by the strong TA increase in the 1990's is exaggerated because of unreliable measurements before 1993. After that, the precision of TA measurements appears to have increased



considerably as is evident from the relatively low scatter in TA values after 1993 as compared to before 1993 (cf. Müller et al. (2016), their Fig. 3). In particular, we find in Fig. 6 that even in the Kattegat deep water the observed TA concentrations in the period ~1985-1992 are comparatively low. While the Kattegat surface waters are heavily influenced by outflowing Baltic Proper water, the Kattegat deep water is affected only very marginally. Furthermore, the observed TA values from the Gotland

Sea deep water in approximately the same period are considerably lower than our modeled values (Fig. 6).

Atmospheric deposition of sulfate, nitrate, and ammonium on the water surface due to emissions from land and ships contribute to a TA decrease and acidification in the Baltic Sea (Hassellöv et al., 2013; Hagens et al., 2014). The impact peaked in the 1980's, but has since then diminished due to reduced land emissions (Omstedt et al., 2015). According to our BALTSEM calculations, the TA sink related to acidic depositions has declined from approximately -40 Gmol y$^{-1}$ in the 1980's to -10 Gmol

y$^{-1}$ in the past decade. This reduced TA sink thus contributes to the increasing TA concentrations in the Baltic Sea.

Riverine TA concentrations can increase as a result of enhanced weathering of carbonate and silicate rocks in the catchments. The rate of weathering depends on temperature, precipitation, soil organic matter contents, and deposition of acids (Ohlson and Anderson, 1990; Dyrssen, 1993; Sun et al., 2017). Average TA loads from Swedish rivers in 1985-2012 and Finnish rivers in 2001-2012 amount to 42 and 15 Gmol y$^{-1}$ respectively (Fig. S2, supplementary material), together corresponding to some

12% of the total TA load (~470 Gmol y$^{-1}$) to the system. There was a strong long-term increase in the flow-normalized TA loads from Swedish rivers – approximately 21% over the period 1985-2012. In Finnish rivers on the other hand, there was no increase in the period 2001-2012. Because of a generally poor availability of river data from other countries around the Baltic Sea, we have no clear understanding of the long-term TA development in e.g. the great rivers in the south-eastern Baltic Sea (where also the highest TA concentrations are generally observed).

Riverine TA concentrations in the BALTSEM model were calculated from observed monthly mean values only in the period 1996-2000. If the long-term increasing trend observed for TA loads in Swedish rivers is also representative for rivers in e.g. the south-eastern Baltic Sea, this would signify that the model is forced by too low riverine TA concentrations in the last decade but on the other hand too high concentrations in the first couple of decades. It is plausible that changes in river water properties are responsible for at least part of the overall increasing TA concentrations in the Baltic Sea. This could in particular

be the case for the Bothnian Sea and Bay where simulated TA in the last decade is underestimated by the BALTSEM model (Fig. 5-6).

### 4.5 Implications

High productivity and deep water oxygen consumption rates favor TA generating anaerobic mineralization processes. One potential consequence is that a large-scale recovery from eutrophication could reduce the $CO_2$ buffering capacity of a marine

system and thus also reduce the atmospheric $CO_2$ sink and surface water pH.

In this section we investigate how the simulated TA in BALTSEM responds to two different nutrient load scenarios: 1. The Business As Usual (BAU) scenario with high nutrient loads throughout the 21$^{st}$ century, and 2. The Baltic Sea Action Plan (BSAP) scenario with large reductions in nitrogen and phosphorus loads (Fig. S3, supplementary material). We use the



ECHAM5 A1B #1 scenario for $CO_2$ emissions and climate change downscaled for the Baltic Sea region (cf. Omstedt et al., 2012). The A1B emission scenario represents a socio-economic development producing medium $CO_2$ emissions where the atmospheric $CO_2$ partial pressure ($pCO_2$) reaches some 700 µatm by the year 2100 (Fig. S4, supplementary material). The unresolved TA source is kept constant throughout these simulations. This means that any simulated changes in TA are related

to changes in river loads and exchange with the North Sea, as well as changes in TA producing/consuming biogeochemical processes that are included in BALTSEM (production, mineralization, denitrification, nitrification, sulfate reduction, sulfide oxidation, etc.).

According to the BALTSEM simulations, the surface and deep water temperatures in the Gotland Sea will increase by approximately 3 degrees over the 21st century, while salinity is reduced by more than 2 (Fig. S5, supplementary material).

Surface water phosphate concentrations will decline by ~0.2 µmol kg$^{-1}$ in the BSAP scenario, resulting in a reduced primary production and increased deep water oxygen concentrations (Fig. S6, supplementary material). The reduced productivity and large-scale recovery from anoxic deep water conditions in the BSAP scenario also will have large consequences for TA and in extension $CO_2$ buffer capacity and pH. Towards the final decades of the simulations, surface water TA in the BAU scenario exceeds that in the BSAP scenario by ~150 µmol kg$^{-1}$ (Fig. 9). As a result, the surface water pH is reduced by 0.1 units more

in the BSAP than in the BAU scenario.

These scenario simulations do not include changes in TA generation resulting from changes in S burial driven by productivity and Fe-oxide availability, since these processes are not resolved in BALTSEM. To investigate how the sediment, and more specifically S formation and burial, will respond to changes in iron and organic carbon loadings, we ran the RTM for an additional 40 years under the present environmental conditions, as well as under a range of changes in these loadings. This

sensitivity analysis (Table 6) shows that reverting the productivity regime to pre-1978 conditions decreases the calculated TA efflux by ~50%, a direct result of less organic matter degradation, whereas S burial is hardly impacted as it is still limited by Fe availability. Lowering the Fe-oxide loading to pre-1973 values decreases the S burial by an order of magnitude, confirming its limitation by Fe. Striking, the TA efflux is only marginally impacted, indicating the decoupling between short-term flux dynamics and long-term TA generation. Increasing the Fe-oxide loading to the peak values of 1981 slightly lowers the TA

efflux while more than doubling S burial, a direct results of a higher Fe availability. In summary, our sensitivity analysis confirms that iron dynamics exert the dominant control on S burial and long-term TA impacts, whereas organic matter input is the major driver of short-term TA effluxes.

It is a simple exercise to examine the sensitivity of pH to further changes in TA. Using the CO2SYS software (Lewis and Wallace, 1998; http://cdiac.esd.ornl.gov/oceans/co2rprtnbk), and assuming that the surface water $pCO_2$ is in equilibrium with

the atmosphere, a surface water $pCO_2$ of 700 µatm and TA of 1425 µmol kg$^{-1}$ (as at the end of the BSAP scenario) results in a surface water pH of 7.77, which is very close to the simulated annual mean pH (Fig. 9). For example, decreasing the surface water TA to 1325 or 1225 µmol kg$^{-1}$ results in pH values of 7.74 or 7.70 respectively. On the contrary, in order to completely compensate for the $CO_2$-induced pH decline resulting from an atmospheric $pCO_2$ increase to ~700 µatm in the A1B scenario, the surface water TA would have to increase to almost 3000 µmol kg$^{-1}$ – which is a completely unrealistic TA concentration



for surface water in the Gotland Sea regardless of productivity and oxygen conditions. It is for that reason beyond any doubt that the only possible way to avoid acidification of Baltic Sea waters is to implement large reductions in $CO_2$ emissions. Although there is a larger pH decline in the BSAP than in the BAU scenario, the possible negative influence must be considered to be of a marginal importance compared to the vast benefits for Baltic Sea ecosystems following reduced deep water dead

zones.

## 5 Summary and concluding remarks

Model calculations have been used to constrain the sedimentary TA efflux in the Baltic Proper, and to examine how this efflux has developed over a 40-year period in relation to eutrophication and oxygen deterioration. In particular, the net TA source related to permanent sulfur burial in the sediment was calculated using a reactive transport model. Furthermore, the physical-

biogeochemical BALTSEM model was used to estimate future TA concentrations and pH levels depending on the development of nutrient loads to the system.

The sedimentary TA generation undergoes large changes depending both on organic matter loads and oxygen conditions. Especially large changes occur during transitions between suboxic and euxinic conditions. Some of these changes are reversible, while others – such as a permanent sulfur burial – result in a net TA generation. Our calculations imply that sulfur

burial in the Baltic Proper has resulted in an average net TA generation of approximately 44 Gmol yr$^{-1}$ in the period 1970-2009. This flux covers ~26% of the missing TA source in this basin (as estimated by the BALTSEM model).

When comparing the BAU and BSAP nutrient loads in combination with the A1B scenario for $CO_2$ emissions, we find a larger pH reduction in the BSAP case than in the BAU case (by approximately 0.1 pH unit). This is a response to reduced signs of eutrophication and particularly substantial improvements in deep water oxygen conditions: In our calculations the gradual

decline in anaerobic mineralization following improved oxygen condition results in a reduced TA generation and thus a reduced buffer capacity for atmospheric $CO_2$ in the Baltic Sea. Sedimentary S burial is not resolved in the BALTSEM model. Additional scenario calculations were for that reason performed with the RTM; the results indicate that S burial and long-term effects on the sedimentary TA efflux are primarily controlled by the iron cycle, while short-term changes in the TA exchange between sediments and the water column mainly depend on organic matter inputs.

**Acknowledgements**

This study was supported by the TRIACID project funded by the Nordic Council of Ministers (Grant #170019) and BONUS COCOA funded by Formas and the European Commission. The Baltic Nest Institute is supported by the Swedish Agency for Marine and Water Management through their grant 1:11 - Measures for marine and water environment. Further funding comes from the Netherlands Organisation for Scientific Research (NWO; Vici 865.13.005 awarded to Caroline P. Slomp) and the

European Research Council under the European Community's Seventh Framework Programme for ERC Starting Grant



#278364. Mathilde Hagens received additional financial support through the Dutch network of Women Professors (LNVH; DWS Fund 2016). We thank the captain and crew of R/V Pelagia (64PE411) for their support, and Matthias Egger, Martijn Hermans and Sharyn Ossebaar for their contributions to the collection of the pore water DIC and TA data. Erik Smedberg is acknowledged for contributions to the artwork.

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





**Tables**

**Table 1. Total sediment areas and muddy sediment areas (1000 km$^2$). The muddy sediment areas are based on Al-Hamdani and Reker (2007). Sub-basins according to Fig. 1.**

| Sub-basins | 1-3 | 4-6 | 7-9 | 10 | 11 | 12 | 13 | 1-13 |
|---|---|---|---|---|---|---|---|---|
| Sediment area | 22.2 | 19.3 | 227.6 | 67.0 | 36.6 | 17.5 | 23.7 | 413.9 |
| Muddy sediment area | 8.5 | 2.3 | 74.3 | 8.8 | 13.3 | 8.7 | 8.8 | 124.5 |





**Table 2. Estimated S burial (mmol m$^{-2}$ y$^{-1}$) and associated TA generation (Gmol y$^{-1}$) for site F80 (58.0000˚N, 19.8968˚E) between 1970 and 2009. Both S contents and dating were taken from Lenz et al. (2015b). The basin scale calculation was based on the total muddy sediment area for the Baltic Proper in BALTSEM of 74300 km$^2$ (Table 1). Numbers between brackets represent S burial due to in situ S formation only, thus excluding S burial due to FeS$_2$ deposition.**

| Source | Sulfur burial (mmol m$^{-2}$ y$^{-1}$) | Total alkalinity generation (Gmol y$^{-1}$) |
|---|---|---|
| observations | 295 | 43.8 |
| model | 291 (164) | 43.2 (24.4) |





**Table 3. Estimated depth-integrated sulfate reduction (SRR), sulfur re-oxidation (S-OX) and $S^0$ disproportionation rates ($S^0$-dispr), as well as $\sum H_2S$ efflux (all in mmol S $m^{-2}$ $y^{-1}$), as derived from the one-dimensional reactive transport model (Reed et al., 2016) for the periods 1970-1973 (baseline; I), 1973-1978 (start change in Fe loading; II), 1978-1981 (eutrophication but pre-euxinia; III) and 1981-2009 (eutrophication and euxinia; IV). The difference between SRR**
5 **(production of reduced S) and the sum of S-OX, $S^0$-dispr and $\sum H_2S$ efflux (removal of reduced S) is assumed to be representative for the maximum potential S formation. The simulated S formation is derived from the mass balance of the RTM (see Table S3 for details).**

| Period | SRR (mmol $m^{-2}$ $y^{-1}$) | S-OX (mmol $m^{-2}$ $y^{-1}$) | $S^0$-dispr (mmol $m^{-2}$ $y^{-1}$) | $\sum H_2S$ efflux (mmol $m^{-2}$ $y^{-1}$) | Potential S formation (mmol $m^{-2}$ $y^{-1}$) | Simulated S formation (mmol $m^{-2}$ $y^{-1}$) |
|---|---|---|---|---|---|---|
| I: 1970-1973 | 1002 | 1.69 | 1.87 | 932 | 66 | 37 |
| II: 1973-1978 | 994 | 1.63 | 1.82 | 845 | 145 | 125 |
| III: 1978-1981 | 1304 | 0.68 | 0.77 | 742 | 561 | 539 |
| IV: 1981-2009 | 2118 | 0.00 | 1.63 | 1942 | 174 | 148 |
| Average | 1805 | 0.42 | 1.61 | 1614 | 189 | 164 |




**Table 4: Estimated TA generation from the reactive transport model integrated over the whole sediment column for the periods 1970-1973 (baseline), 1973-1978 (start change in Fe loading), 1978-1981 (eutrophication but pre-euxinia) and 1981-2009 (eutrophication and euxinia) for the dominant processes, as well as TA generation and efflux (all in mmol m$^{-2}$ y$^{-1}$).**

*Primary reactions (all in mmol m$^{-2}$ y$^{-1}$)*

| Period | $OM + NO_3^-$ | $OM + MnO_2$ | $OM + Fe(OH)_3$ | $OM + SO_4^{2-}$ | $OM + CH_4$ |
|---|---|---|---|---|---|
| 1970-1973 | 36 | 21 | 130 | 966 | 24 |
| 1973-1978 | 33 | 20 | 361 | 946 | 24 |
| 1978-1981 | 23 | 9 | 1764 | 1547 | 36 |
| 1981-2009 | 0 | 1 | 51 | 2706 | 103 |
| Average | 9 | 5.67 | 226 | 2225 | 80 |

*Secondary reactions (all in mmol m$^{-2}$ y$^{-1}$)*

| Period | $SO_4 + CH_4$ | $O_2 + Fe^{2+}$ | $Fe^{2+} + \sum H_2S$ | $Fe(OH)_3 + \sum H_2S$ | $O_2 + \sum NH_4^+$ |
|---|---|---|---|---|---|
| 1970-1973 | 1125 | -115 | -37 | 35 | -145 |
| 1973-1978 | 1125 | -307 | -131 | 110 | -135 |
| 1978-1981 | 1199 | -1175 | -567 | 330 | -69 |
| 1981-2009 | 1770 | 0 | -146 | 184 | 0 |
| Average | 1582 | -138 | -165 | 170 | -36 |

*Total (all in mmol m$^{-2}$ y$^{-1}$)*

| Period | Total OM degradation | Total secondary reactions | Total TA generation | Modeled TA efflux |
|---|---|---|---|---|
| 1970-1973 | 1178 | 836 | 2015 | 1901 |
| 1973-1978 | 1385 | 636 | 2022 | 1683 |
| 1978-1981 | 3381 | -300 | 3081 | 2361 |
| 1981-2009 | 2861 | 1800 | 4661 | 4423 |
| Average | 2547 | 1401 | 3948 | 3674 |



**Table 5. Resolved and unresolved TA sources minus sinks (SMS) and river loads (Gmol y$^{-1}$) in 1970-2014 according to the BALTSEM calculations in this study. Sub-basins according to Fig. 1.**

| Sub-basins | 1-3 | 4-6 | 7-9 | 10 | 11 | 12 | 13 | 1-13 |
|---|---|---|---|---|---|---|---|---|
| Resolved pelagic SMS | 8.2 | 10.2 | 60.4 | 8.9 | -0.4 | 8.8 | 9.8 | 105.9 |
| Resolved benthic SMS | -2.7 | -2.6 | 29.3 | -4.7 | -1.0 | -3.1 | -1.7 | 13.5 |
| Total resolved SMS | 5.5 | 7.5 | 89.7 | 4.2 | -1.4 | 5.7 | 8.2 | 119.4 |
| Unresolved SMS | - | - | 166.2 | 24.5 | 6.7 | 25.6 | 34.6 | 257.5 |
| Total SMS | 5.5 | 7.5 | 255.9 | 28.7 | 5.3 | 31.3 | 42.8 | 377.0 |
| River load | 21.0 | 13.5 | 216.8 | 25.6 | 16.7 | 95.8 | 80.1 | 469.5 |




**Table 6. Input of Fe-oxides and simulated S burial and TA efflux averaged for the period 2011-2050 under a range of environmental conditions as calculated with the reactive transport model. All values are in mmol $m^{-2}$ $y^{-1}$.**

| Scenario | Input of Fe-oxides (mmol $m^{-2}$ $y^{-1}$) | Sulfur burial (mmol $m^{-2}$ $y^{-1}$) | TA efflux (mmol $m^{-2}$ $y^{-1}$) |
|---|---|---|---|
| business-as-usual | 60 | 235 | 4910 |
| no eutrophication | 60 | 232 | 2558 |
| pre-1973 Fe loading | 20 | 23 | 4919 |
| peak Fe loading (1981) | 360 | 534 | 4816 |



**Figures**

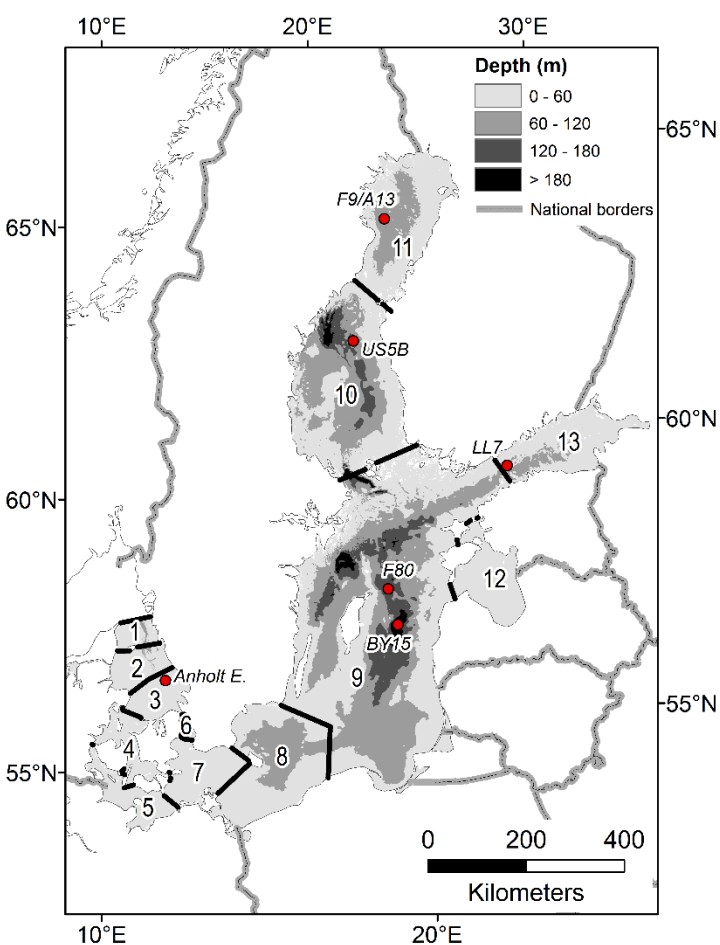

Figure 1: . The Baltic Sea area with sub-basins and monitoring stations. The sub-basins are: 1. Northern Kattegat (NK), 2. Central Kattegat (CK), 3. Southern Kattegat (SK), 4. Samsø Belt (SB), 5. Fehmarn Belt (FB), 6. Öresund (OS), 7. Arkona Basin (AR), 8. Bornholm Basin (BN), 9. Gotland Sea (GS), 10. Bothnian Sea (BS), 11. Bothnian Bay (BB), 12. Gulf of Riga (GR), 13. Gulf of Finland (GF).



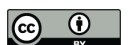

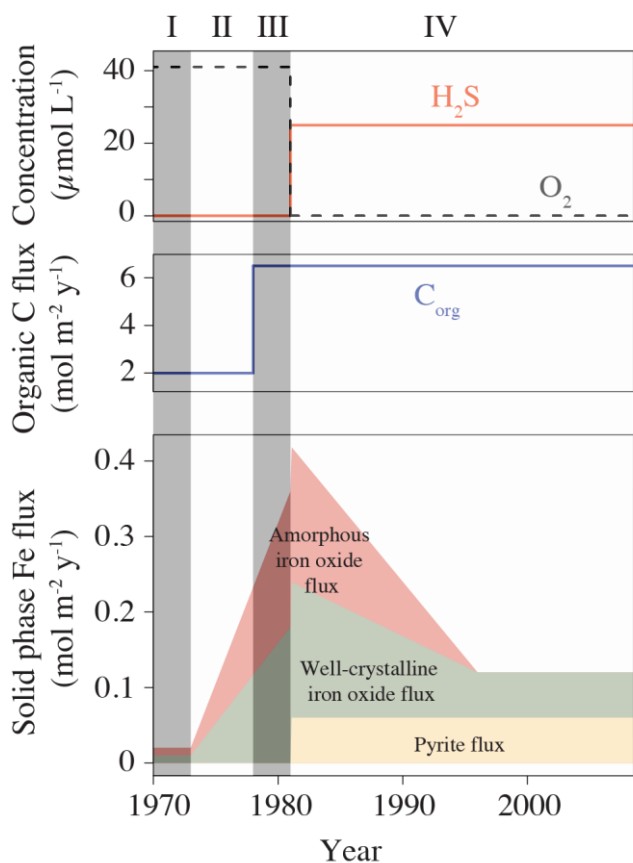

**Figure 2: Variability in bottom-water redox conditions, organic carbon and iron inputs between 1970-2009, used to force the reactive transport model at site F80. Numbers indicate the four different time intervals recognized in this study: I. baseline (1970-1973); II. start of change in Fe loading (1973-1978); III. eutrophication but pre-euxinia (1978-1981); IV. eutrophication and euxinia (1981-2009). Figure modified from Reed et al. (2016).**



**Figure 3: Pore water profiles of selected simulated (solid lines) and observed (dots) variables at site F80, as well as simulated rates of some major processes impacting TA dynamics (all rates in mmol TA dm$^{-3}$ y$^{-1}$). Additional pore water and solid phase profiles were published in Reed et al. (2016). Previously unpublished measurements can be found in Table S2 (supplementary material).**





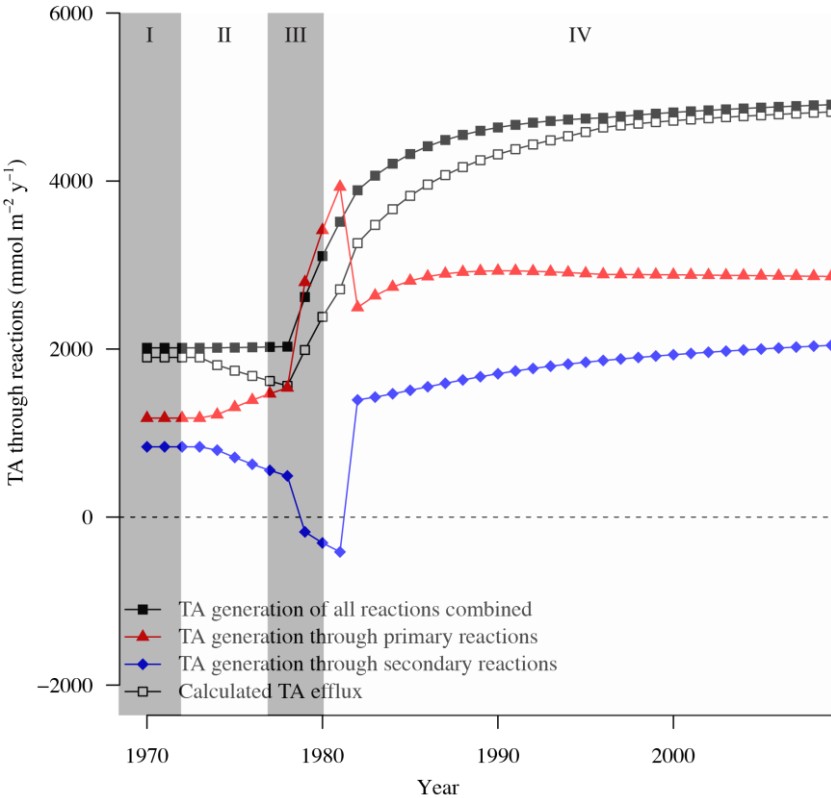

**Figure 4: Calculated TA efflux and generation at site F80 integrated over the whole sediment column (all in mmol m⁻² d⁻¹) due to various biogeochemical processes implemented in the RTM.**




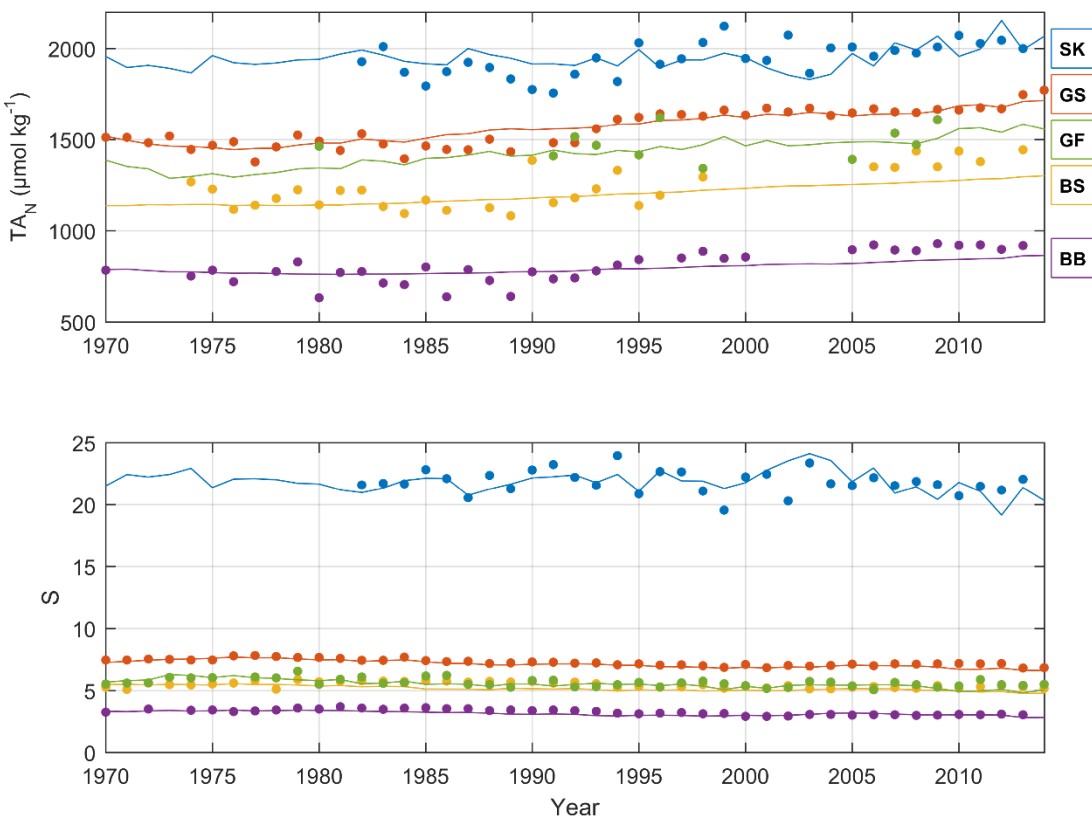

**Figure 5: Annual mean observed (dots) and modeled (lines) normalized surface water TA (TA$_N$, μmol kg$^{-1}$) and salinity in five sub-basins. Model data from sub-basin 3 (SK), 9 (GS), 10 (BS), 11 (BB), and 13 (GF) are compared to observed data at the Anholt East, BY15, US5B, F9/A13, and LL7 stations respectively.**



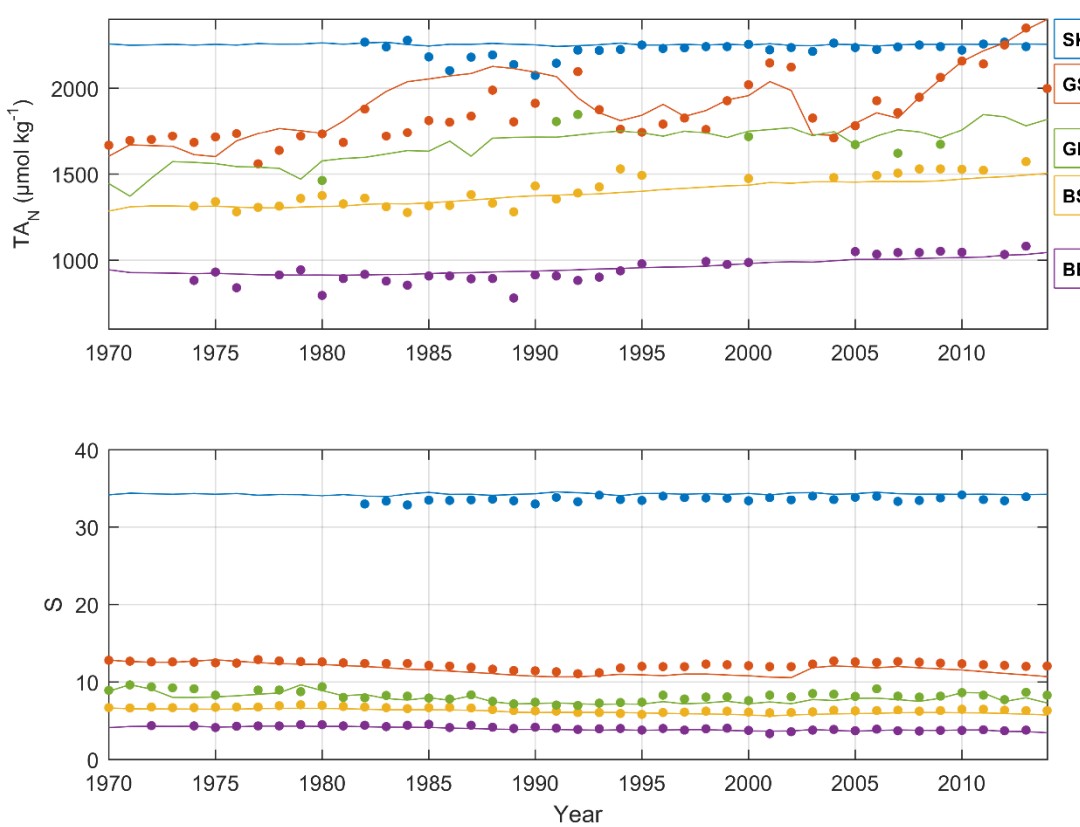

**Figure 6: Annual mean observed (dots) and modeled (lines) normalized deep water TA (TA$_N$, μmol kg$^{-1}$) and salinity in five sub-basins. Model data from sub-basin 3 (SK), 9 (GS), 10 (BS), 11 (BB), and 13 (GF) are compared to observed data at the Anholt East, BY15, US5B, F9/A13, and LL7 stations respectively.**




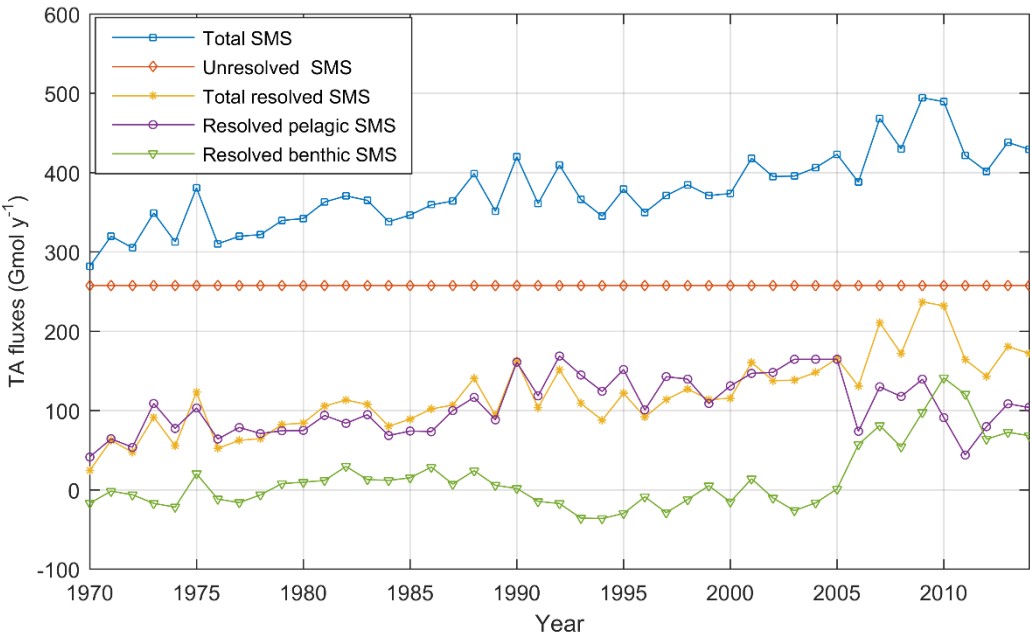

**Figure 7: Annual mean TA sources minus sinks (SMS) (Gmol y⁻¹) in the entire Baltic Sea according to BALTSEM calculations: Resolved benthic sources minus sinks (SMS) (Gmol y⁻¹) (green line), resolved pelagic SMS (purple line), total resolved SMS (yellow line), unresolved SMS (red line), and total SMS (blue line).**





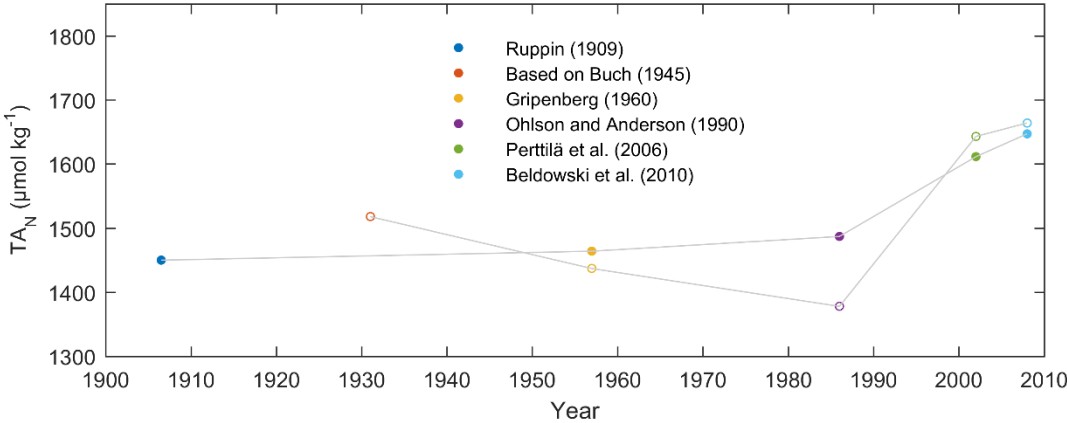

**Figure 8: TA concentrations normalized to salinity = 7 (TA$_N$), using the TA-salinity relations for the Baltic Proper – Kattegat (full circles), and Baltic Proper – Gulf of Bothnia (open circles) (cf. Table S4-S5, supplementary material). The Ruppin (1909) value was based on measurements in 1906-1907 (see Dyrssen (1993) for reference), while the Buch (1945) value was based on measurements in 1927-1935 (cf. Buch, 1945). The Buch (1945) and Perttilä (2006) values were converted to µmol kg$^{-1}$ from µmol l$^{-1}$.**





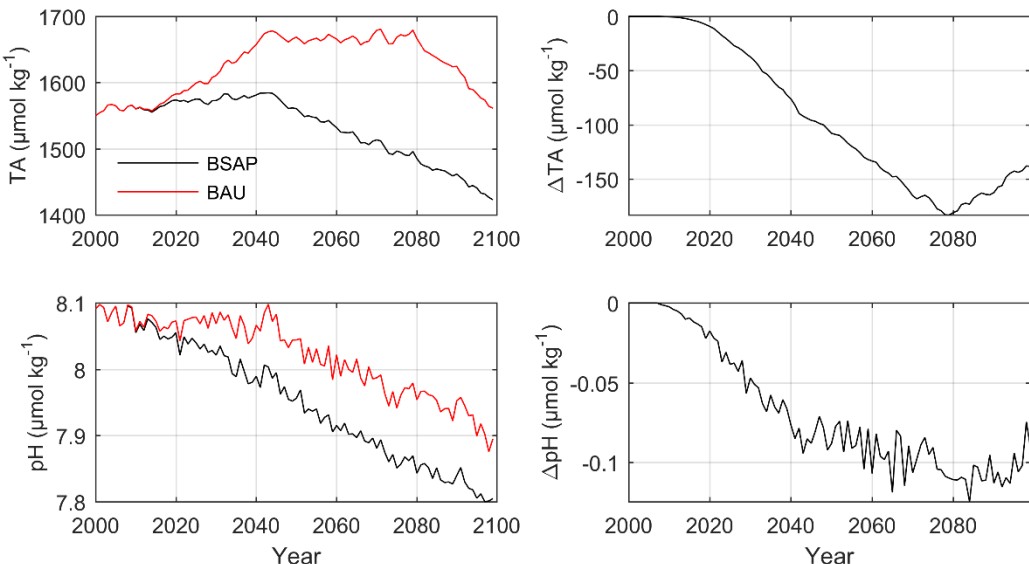

**Figure 9: Left: simulated annual mean surface water TA and pH in the Gotland Sea according to the BSAP (black lines) and BAU (red lines) nutrient load scenarios respectively. Right: differences between the BSAP and BAU scenarios.**