# Peer review of "Sedimentary alkalinity generation and long-term alkalinity development in the Baltic Sea"

_Biogeosciences, 2018_

## Referee Comment (RC1) · Anonymous Referee #1 · 7 Aug 2018

This manuscript reports alkalinity development in the Baltic Sea since 1970 based on two models. I am, however, not convinced about the results as there is a lot of extrapolation, and the calculations are done without an error analysis. For instance, it is said that 260Gmol/y of the additional TA source can not be explained but there is no error bar. In fact, most of the numbers given have no error bars. My other major concerns are: 1. Sulfur in umol/g is converted to umol/cm3 using measured porosity but this can not be done using porosity alone. 2. The results are integrated to 25cm and extrapolated to basin scale. But, how reliable is the age model and how homogeneous is the sediments in the Baltic Sea? I would expect a lot of variability in both but the uncertainty is not evaluated. 3. It is said that borate is ignored because it is expected to have a low contribution to TA. I am surprised. What is the basis for

such a statement? My minor concern is the statement that the Baltic Sea today forms the largest anthropogenic dead zone is the world. It is not true. There are many dead zones in the Baltic Sea so the authors must have summed them up. If one sums up separate dead zones in other seas, such as the East China Sea, the total area is much larger. In fact, the dead zone off the Changjiang river mouth alone is now larger.

---

## Referee Comment (RC2) · Anonymous Referee #2 · 13 Aug 2018

Overall Statements

The manuscript "Sedimentary alkalinity generation and long-term alkalinity development in the Baltic Sea" by Erik Gustafsson and colleagues presents the simulated development of alkalinity generation in the Baltic Sea over the last decades and, additionally, projections until 2100. The modelling tools include a reactive-transport model (RTM) for sedimentary processes which is able to resolve Fe-S cycling and burial of corresponding components, which in turn generates TA. Such irreversible processes are necessary to describe the missing (unresolved) contributors to the overall TA sources in the Baltic Sea.

Instead of a coupled physical – biogeochemical 3D model which couples benthic and pelagic processes, the authors use the less expensive model BALTSEM for the different Baltic Sea basins and the RTM which is weakly coupled to BALTSEM. For the reader it is unclear which information (fluxes) are provided for RTM by BALTSEM and vice versa. A full bidirectional coupling of both models, which is claimed as not feasible (I doubt) will definitively produce results differing from the weak applied coupling. It is necessary to estimate the error induced by this weak coupling. I suggest to test this with an application of BALTSEM for one water column and the underlying RTM sediment core. Within one scenario the weak coupling should be applied and within another scenario a full coupling should run. With these two results the authors can compare the TA generation of both scenarios and hopefully are able to demonstrate that the result of the weak coupling shows the main TA-related features as the full coupled run.

One of the main conclusions of the manuscript is that Fe-S dynamics impact the TA generation only on longer time scales. This is derived from one sentence on page 15 line 23. For this conclusion I expect a deeper analysis.

Detailed remarks

P2 L3: Sarmiento and Gruber, 2006: Ref missing

P2 L8: Rabalais et al., 2015: Ref says 2014

P2 L17 and L25 Reference List shows only Hu and Cai, 2011

P3 L1: Table 1 in Gustafsson et al 2014b gives 453 Gmol yr$^{-1}$ as riverine TA load.

P4 L21: The expression $\Sigma H_2 S$ must be introduced.

P5 L2: How large was the increase of TA loads when the new Swedish and Finnish data were included?

P5 L17: Lukawska-Matuszewska and Kielczewska, 2016

P6 L18: The use of these unresolved fluxes is very unsatisfying. They might also represent sinks that are assumed too high. Using such a "joker", it's relatively easy to match observed TA concentrations.

P7 L3ff: How do you handle the lateral Fe input? How do you treat S burial and the consecutive TA flux into the pelagic? The normal way across pore water diffusion in connection with overlying water cannot work with this model setup.

P7 L10 Describe the upscaling process in more detail.

P7 L19 I do not see the problem to handle 1400 sediment "boxes".

P7 L22 You should say that the current model setup is only an intermediate step towards full coupling.

P10 L14-20 The text is non-transparent. Enumerate all shortcuts and discuss their implications. Specify the processes and species, which cannot be linked.  Here, the above mentioned sensitivity study should be discussed.

P12 L24 Dijkstra et al,. 2018: Ref says 2017

P15 L23: "Striking .." Discuss this item in more detail. Why would you have assumed a stronger impact? Which mechanism hampers it?

P19 L6: "2014a"

P19 L9: "2014b"

P36 L3: Ruppin (1909): Ref missing

---

## Referee Comment (RC3) · Anonymous Referee #3 · 19 Aug 2018

Gustafsson et al. investigated benthic TA generation under low oxygen conditions using a coupled physical-biogeochemical model BALTSEM and a reactive-transport sediment model (RTM). They discovered that Fe sulfide burial accounts for roughly a quarter of "missing" TA that cannot be accounted for using the BALTSEM model, with the rest of the missing TA potentially coming from groundwater and river input. In addition, the authors found that Fe limitation restricts the magnitude of the Fe-S burial produced TA. Using scenario modeling, this anaerobic TA source will become even smaller because of improvement of oxygen conditions and reduced Fe input from land, hence burial of reduced sulfur. As a result, the Baltic Sea will be more prone to the acidification risks.

The major concern I have is how representative the RTM model result is across the entire Baltic Sea. The authors stated that this model calculation is very expensive if multiple stations are included. It appears that they tried to extrapolate only one station (F80) to the entire Baltic basin where this is "muddy" sediments. It is known that sediments are heterogeneous and the Baltic definitely should not be an exception. Therefore, I question the validity of their data presentation and interpretation based on the RTM model.

The authors also admitted that historical TA measurement in rivers may have bias due to the lack of modern analytical standard (such as CRM) and approaches. If they can add a section of error analysis and see how that could contribute to the "missing" TA in the BALTSEM model. Given that groundwater and river could both contribute large fraction of TA budget in this area and potentially changing hydrological conditions, a lack of this analysis make the entire argument in the manuscript a bit flawed.

I would also add a table on the types of reactions involved in both the BALTSEM and RTM models. The authors discussed them in multiple places but sometime they were rambling, which makes it hard to follow.

Scientific questions:

P3 L2-L8, the background on TA budget is not clear. First Gustafsson et al. stated that there is an additional TA source of 344 Gmol/yr is needed to close the Baltic Sea TA budget, and 260 Gmol/yr cannot be explained, while 84 Gmol/yr was resolved with 18 Gmol/yr was attributed to net sulfate reduction. At this point, it seems that the problem has been resolved (net sulfate reduction). Then on L7, it seems that the authors separated FeS burial from net sulfate reduction, although FeS involves net sulfate reduction. Therefore, these budgetary terms need better explanation.

P6 L16-30, the argument is weak. If the alkalinity contribution isn't known well (L28), how would the authors feel comfortable to run the calibration and ascertain the direction of sedimentary TA flux?

P8 L23-25, First if you refer to a multiple panel figure, please try to using letters (a, b, c...) to distinguish these panels. More importantly, if sulfide is being produced, according to reaction stoichiometry the authors presented, it would lead to TA accumulation, regardless whether this sulfide goes down or not.

P9 L8-11, the statement that maximum CH4-driven sulfate reduction at 8 cm is the greatest is interesting and need better explanation. If you show sulfate profile, this may look like where the sulfate-methane interface (SMZ) is and may have nothing to do with the temporal variations of CH4 production.

P9 L12 paragraph, the authors separated different depth layers to account for the reactions that have TA implications. However, they failed to show whether TA produced in these layers make it up to TA efflux to the water column. This paragraph needs better organization and clarification.

Technical comments:

Indentation of paragraphs would be useful

P2 L20-L25, these sentences are a bit confusing. "On a system scale" is mentioned twice (L20 and L24). The two examples after "For examples" are essentially the same thing, so it's not clear why the authors used "however" to connect the two sentences.

P3 L3, "unresolved" TA is 260 Gmol/yr, while in P9 L22 this number is 257.5 Gmol/yr, please be consistent.

P3 L20, the two paragraphs need a better transition. Elemental sulfur seems to be introduced into the context suddenly. Prior to doing so, the authors need to explain a bit of the type of reduced sulfur burial, i.e., sulfide (with Fe) and elemental sulfur. Relative abundance of these two types of reduced sulfur also needs to be presented based on the literature.

P4 L1, is it integrated to 25 cm "interval"?

P4 L6-L15, this should go into the Introduction, as really this should be background information, by no means materials and methods.

P4 L24, move "for that reason" to the beginning of the sentence.

P4 L25, where exactly does sigma-$H_2S$ come from?

P5 L5 paragraph, while the authors referred to a supplemental table in a published study, it would be desirable to include such information in their own supplemental materials and refer to it in the context. Otherwise the readers can easily get lost.

P5 L17, not a complete sentence.

P7 Line 12, what is "This" approach?

P7 L25 and P8 L10, why are the two sulfur burial in the same time period (1970-2009)

different?

P11 L9, where is 166 Gmol/yr in Table 5?

P11 L29 and P9 L 8, I believe 43% and 43.8% in these two places are the same thing, please be consistent, at least don't round off the number in the wrong way.

P11 Section 4.2, both unit TA (mmol/m2/yr) and overall TA flux (Gmol/yr) are presented, I'd suggest that you stick with one or both (with parentheses) to avoid confusion and don't let the readers to do the conversion.

P12 L1 and P11 L31, this sounds like freshwater environment but not typical marine since sulfate is depleted but there is still oxidized Fe.

P14 L6, not precise, it's the deposition of SOx and NOx that contribute to TA reduction, ammonia deposition first increase TA and decrease upon oxidation (or biological uptake), if no biological effect is involved, there is no TA implication, i.e., deposition followed by oxidation or uptake.

P14, the title of Section 4.5, this is not implications, rather simulations of future scenarios.

P15 L29, please update the link.

P15 L33-34, what about including possible temperature increase? It would be a better approach to simulate future pH with invariant TA, like in most studies.

---

## Author Comment (AC3) · 25 Sep 2018

Gustafsson et al. investigated benthic TA generation under low oxygen conditions using a coupled physical-biogeochemical model BALTSEM and a reactive-transport sediment model (RTM). They discovered that Fe sulfide burial accounts for roughly a quarter of "missing" TA that cannot be accounted for using the BALTSEM model, with the rest of the missing TA potentially coming from groundwater and river input. In addition, the authors found that Fe limitation restricts the magnitude of the Fe-S burial produced TA. Using scenario modeling, this anaerobic TA source will become even smaller because of improvement of oxygen conditions and reduced Fe input from land, hence burial of reduced sulfur. As a result, the Baltic Sea will be more prone to the acidification risks. The major concern I have is how representative the RTM model result

is across the entire Baltic Sea. The authors stated that this model calculation is very expensive if multiple stations are included. It appears that they tried to extrapolate only one station (F80) to the entire Baltic basin where this is "muddy" sediments. It is known that sediments are heterogeneous and the Baltic definitely should not be an exception. Therefore, I question the validity of their data presentation and interpretation based on the RTM model.

Response: The calculated fluxes at F80 were not extrapolated to the muddy areas of the entire Baltic Sea. Instead, we use it to represent muddy areas in the Baltic Proper (or central Baltic Sea) which is where F80 is located. However, there are indeed spatial differences in the sediment geochemistry of muddy Baltic Proper sediments. To clarify the uncertainties and limitations with our approach, the following paragraph has been added to Section 4.1: "The RTM fluxes are upscaled under the assumption that the fluxes computed for the F80 site are representative for the muddy sediment area of the Baltic Proper. This assumption is associated with uncertainties because of spatial differences in the sediment geochemistry of muddy Baltic Proper sediments as illustrated by the porewater and Fe-S chemistry for 4 other sites as published by Lenz et al. (2015). The solid phase profiles for these sites show similar temporal trends over the past decades as F80. Furthermore, the porewater profiles show that site F80 has a relatively high rate of organic matter deposition and alkalinity regeneration when compared most of the other sites. This implies that, with our extrapolation, the role of the sediment could be slightly overestimated. Thus, the large-scale fluxes we obtain by extrapolating fluxes from one specific site are to be regarded as a high-end estimate of e.g. the possible contribution of Fe-S burial to the overall TA budget".

The authors also admitted that historical TA measurement in rivers may have bias due to the lack of modern analytical standard (such as CRM) and approaches. If they can add a section of error analysis and see how that could contribute to the "missing" TA in the BALTSEM model. Given that groundwater and river could both contribute large fraction of TA budget in this area and potentially changing hydrological conditions, a

lack of this analysis make the entire argument in the manuscript a bit flawed.

Response: It is mainly the historical measurements in the Baltic Sea (and not in the rivers) that are unreliable – at least in certain periods. This is partly due to methods, and (probably more important) partly due to handling of the water samples. For example, many measurements from the 1980's and early 1990's are very likely flawed (as discussed in Section 4.4). The simulated river loads of TA are (as described in Section 2.1.3) based on measurements in the period 1996-2000 where we have data from most of the major rivers entering the Baltic. The reason that we use only this short period is that it is a major obstacle to achieve data from many of the large (and alkalinity-rich) continental rivers (whereas we have long time-series form Swedish and Finnish alkalinity-poor rivers). It is furthermore difficult to judge the quality of measurements in some of these rivers but we don't know how large the error might be and how much of the missing source that can be resolved if it were possible to reliably update the river loads. One main purpose of this paper is to estimate how much sedimentary processes can contribute to the overall TA budget of the Baltic Sea – and in particular to what extent these processes can explain the missing link. Trying to improve the river loads has not been a goal although this is something that we very much would like to do in future studies. But again, the bottleneck here is really to achieve (high-quality) data from all major rivers in the first place. Nevertheless, to clarify a bit of the uncertainty related to the parameterization of the unresolved sources, we added a second BALTSEM simulation where the unresolved sources in the various sub-basins are added as land loads instead of sediment release. Thus, the simulations now cover the two extreme cases where the unresolved source is either explained by land loads or by sediment release. We did however not want to speculate about how these contributions might vary between the different basins (see further in comment below).

I would also add a table on the types of reactions involved in both the BALTSEM and RTM models. The authors discussed them in multiple places but sometime they were rambling, which makes it hard to follow.

Response: We agree. In the current version, it may not be so easy for the reader to keep track of what is included in the different models and what is not included (without consulting prior studies). We have now added three new tables to the supplementary material: In Table S1-S2 we list primary and secondary redox reactions included in the RTM, while in Table S3 we list the TA influencing reactions included in BALTSEM. This will now also be clearly indicated in Section 2.2.1 and Section 2.2.2.

Scientific questions:

P3 L2-L8, the background on TA budget is not clear. First Gustafsson et al. stated that there is an additional TA source of 344 Gmol/yr is needed to close the Baltic Sea TA budget, and 260 Gmol/yr cannot be explained, while 84 Gmol/yr was resolved with 18 Gmol/yr was attributed to net sulfate reduction. At this point, it seems that the problem has been resolved (net sulfate reduction). Then on L7, it seems that the authors separated FeS burial from net sulfate reduction, although FeS involves net sulfate reduction. Therefore, these budgetary terms need better explanation.

Response: BALTSEM calculates the net sulfate reduction (sulfate reduction minus sulfide oxidation), but this TA source is completely reversible depending on oxygen conditions. Fe-S burial is not included in BALTSEM because we are not presently capable of modelling the coupled Fe-S cycling in the sediments (Fe is not included in the model). Thus, Fe-S burial is a TA source that we can't explicitly take into account in BALTSEM, and this is why we need the RTM where the sedimentary processes are resolved in detail. We updated the text in Section 1: "The remaining 18 Gmol y-1 resulted from net sulfate reduction (sulfate reduction – sulfide oxidation) in the water column, but this fraction could be reversed in case of oxygenation of the water column. It was hypothesized that a significant fraction of the unresolved TA source could be coupled to burial of Fe sulfides as a result of anaerobic mineralization in sediments. This would then represent a fraction of the sulfate reduction that is not readily reversed upon re-oxygenation of the water column.".

P6 L16-30, the argument is weak. If the alkalinity contribution isn't known well (L28), how would the authors feel comfortable to run the calibration and ascertain the direction of sedimentary TA flux?

Response: As described above, we have added a second BALTSEM simulation where the unresolved sources are instead added as land loads to the different sub-basins. The following paragraph has been added to Section 2.2.2: "The processes behind the unresolved TA sources are not known, but there are two candidates: external loads (e.g. river loads and submarine groundwater discharge) and internal processes (pelagic and/or benthic). Rather than speculating about contributions from various sources in the different sub-basins, we will perform two different scenarios: one case were all unresolved sources are added as additional land loads, and one case were the sources are added as sediment release". The resulting TA concentrations in different sub-basins are added as additional lines in Fig. 5-6, and the following paragraph is included in Section 4.2: "In the two different scenarios where the unresolved source is added as land loads (full lines in Fig. 5-6) or sediment release (dashed lines in Fig. 5-6), the simulated surface water TA concentrations are very similar (Fig. 5). Deep water concentrations on the other hand differ significantly in the Gotland Sea and the Gulf of Finland but not in the other sub-basins (Fig. 6). The reason behind the rather similar results for these two different scenarios is that land loads supplied to the different basins are rapidly distributed in the well mixed surface layer, and the well mixed surface layer constitutes a large majority of the water volume. In the deeper and more isolated parts of the system, TA concentrations are considerably lower in the "land loads" case compared to the "sediment release" case. This is a clear indication of the need for an additional TA source in the deeper basins".

P8 L23-25, First if you refer to a multiple panel figure, please try to using letters (a, b, c...) to distinguish these panels. More importantly, if sulfide is being produced, according to reaction stoichiometry the authors presented, it would lead to TA accumulation, regardless whether this sulfide goes down or not.

Response: In the revised manuscript we will add letters to the sub-plots and corresponding parts of the text when the figure is discussed. The other point that the reviewer raises here is an excellent point. The argument on P8 L23-25 refers to the fact that the model overestimates the buildup of 2S. Indeed, the production of sulfide leads to a generation of TA if this sulfide is not reoxidized, as is the case in the current model simulation where the sulfide is being built up in the porewater (which we argue is an artefact resulting from the chosen lower boundary condition for 2S). We cannot assess the fate of the sulfide when it would diffuse downwards below 32 cm depth, as we suggest that actually happens. This makes it difficult to link this 2S production to a permanent, long-term TA source. However, we agree that it is not likely that the 2S will be reoxidized in the time period of interest. This indicates that the net TA source to the water column due to sulfide oxidation would be higher. In the revised manuscript, we will address this and quantify the additional TA generation due to the buildup of 2S in the porewater.

P9 L8-11, the statement that maximum CH4-driven sulfate reduction at 8 cm is the greatest is interesting and need better explanation. If you show sulfate profile, this may look like where the sulfate-methane interface (SMZ) is and may have nothing to do with the temporal variations of CH4 production.

Response: This is a valid point raised by the reviewer. The percentage of CH4-driven sulfate reduction due to upward diffusing CH4, which is on average ∼95% for the period 1970-2009, does change over time (on average ∼98, ∼98, ∼98 and ∼94% for the four periods 1970-1973, 1973-1978, 1978-1981 and 1981-2009, respectively), but this change is indeed relatively minor, and upward diffusing CH4 still dominates. So, we agree that the change in position of the SMTZ is more important here. The temporal evolution of porewater profiles of SO4 and CH4 are presented in Reed et al. (2016, their Fig. 6). From this it follows that the position of the SMTZ is shifting upwards over time, especially since 1981, when Fe(OH)3 and O2 declined as important terminal electron acceptors. Therefore, more SO4 was consumed by OM degradation, and the

SMTZ shifted upwards. We will address this in the revised manuscript by replacing the NH4 and PO4 plots in Fig. 3 by the SO4 and CH4 plots, adding the location of the SMTZ for the various time periods, and altering the text to include the above discussion.

P9 L12 paragraph, the authors separated different depth layers to account for the reactions that have TA implications. However, they failed to show whether TA produced in these layers make it up to TA efflux to the water column. This paragraph needs better organization and clarification.

Response: Throughout the manuscript, we argue extensively that TA effluxes are not representative for answering the research question that we are investigating (e.g. section 2.2.3, P8 L30ff, section 4.1, P15 L23ff). We therefore decided not to discuss the TA efflux in too much detail. The results discussed in the paragraph on P9 L12ff indicate, similarly to P15 L23ff, that processes that impact the TA efflux are not the same processes that impact S burial or total TA generation, but that the sediment depth at which processes take place also plays a role here. In the revised manuscript we will further stress this message.

Technical comments:

Indentation of paragraphs would be useful

Response: Indentations are now added.

P2 L20-L25, these sentences are a bit confusing. "On a system scale" is mentioned twice (L20 and L24). The two examples after "For examples" are essentially the same thing, so it's not clear why the authors used "however" to connect the two sentences.

Response: This section is now rewritten, and the paragraph reads as follows: "The ultimate buildup of TA due to primary production and mineralization depends on the source of the reactants and/or the fate of the products of all alkalinity-generating/consuming reactions. For example, production of dinitrogen gas (N2) during pelagic or benthic denitrification results in a permanent loss of nitrate and hence a gain of TA (Soetaert

et al., 2007). On a system scale this process only results in net TA production if the nitrate is derived from an external source rather than from local nitrification (Hu and Cai, 2011b). Similarly, sulfate reduction leads to net TA generation only if the produced sulfide is buried as e.g. iron (Fe) sulfides rather than being reoxidized within the same system (Hu and Cai, 2011a). Note that the location of sulfide reoxidation, i.e. sediment or water column, impacts net TA generation in the sediments but not on a system scale".

P3 L3, "unresolved" TA is 260 Gmol/yr, while in P9 L22 this number is 257.5 Gmol/yr, please be consistent.

Response: Corrected.

P3 L20, the two paragraphs need a better transition. Elemental sulfur seems to be introduced into the context suddenly. Prior to doing so, the authors need to explain a bit of the type of reduced sulfur burial, i.e., sulfide (with Fe) and elemental sulfur. Relative abundance of these two types of reduced sulfur also needs to be presented based on the literature.

Response: With S we indicate (reduced) solid sulfur in any form, not only elemental sulfur. For the rest, this comment ties in with the comment below, where it is suggested to move the text of P4 L6ff to the introduction. When doing so, we directly explain why for this work it is not relevant in which form the elemental sulfur is present.

P4 L1, is it integrated to 25 cm "interval"?

Response: Yes, we mean the interval between 0 and 25 cm sediment depth. This will be rewritten.

P4 L6-L15, this should go into the Introduction, as really this should be background information, by no means materials and methods.

Response: We partly agree with the reviewer here, and in the revised manuscript most of this text will be moved to the introduction. However, we are aware that various

papers use various reaction equations and stoichiometries to link sulfur burial / pyrite production, mostly depending on the temporal and / or spatial scale of investigation. For example, in Łukawska-Matuszewska and Graca (2018) a different reaction equation was used for this. Therefore, we think it is important to explain the rationale behind our choice of reaction equations, and the materials and methods seems like a proper place to do so.

P4 L24, move "for that reason" to the beginning of the sentence.

Response: Ok.

P4 L25, where exactly does sigma-H2S come from?

Response: This is now explained in Section 2.1.2 (2S = [HS-] + [H2S]).

P5 L5 paragraph, while the authors referred to a supplemental table in a published study, it would be desirable to include such information in their own supplemental materials and refer to it in the context. Otherwise the readers can easily get lost.

Response: Reactions included in the RTM and BALTSEM respectively are now described in Table S1-2 (RTM) and Table S3 (BALTSEM), respectively (see comment above).

P5 L17, not a complete sentence.

Response: We are not sure which sentence this refers to; this section seems fine.

P7 Line 12, what is "This" approach?

Response: This is now rewritten.

P7 L25 and P8 L10, why are the two sulfur burial in the same time period (1970-2009) different?

Response: Part of the sulfur solids is derived from settling of water-column particles onto the sediment surface. This cannot be derived by looking at the S solids concentrations alone (P7 L25), but becomes clear when investigating the inputs of FeS2 to the sediment (Figure 2) and in-situ reaction rates (Table S3; P8 L10), as is done with the RTM. For the net TA generation, however, the total S burial is most important, because the burial prevents possible re-oxidation in either sediment or water column and thus represents the long-term TA source that we are interested in.

P11 L9, where is 166 Gmol/yr in Table 5?

Response: It is there, sub-basin 7-9. The description of the division into different sub-basins and larger areas has been improved in the text. This is now described in the caption of Fig. 1 and also in the text (Section 2.1.1 and Section 4.2). The legends of Table 1 and 5 have been updated as well for clarity.

P11 L29 and P9 L 8, I believe 43% and 43.8% in these two places are the same thing, please be consistent, at least don't round off the number in the wrong way.

Response: This is correct and must have been a typo, as the correct (non-rounded) percentage is 43.3%. We will use ∼43% on both occasions.

P11 Section 4.2, both unit TA (mmol/m2/yr) and overall TA flux (Gmol/yr) are presented, I'd suggest that you stick with one or both (with parentheses) to avoid confusion and don't let the readers to do the conversion.

Response: Throughout the manuscript, we stick to mmol/m2/y when discussing the sedimentary fluxes. Only when we link the results to BALTSEM, like in this paragraph, we convert them to Gmol/y.

P12 L1 and P11 L31, this sounds like freshwater environment but not typical marine since sulfate is depleted but there is still oxidized Fe.

Response: Iron-mediated anaerobic oxidation of methane has found to be significant in deeper Baltic Sea sediments that were subjected to a lake-marine transition (Egger et al., 2017), as well as in coastal sediments of the Bothnian Sea, which has low salinity (Rooze et al., 2016). Indeed, these conditions are not typical marine, so discussing

this process is not too relevant in the context of this manuscript. We included it to be as complete as possible, but will remove it from the revised version of the manuscript.

P14 L6, not precise, it's the deposition of SOx and NOx that contribute to TA reduction, ammonia deposition first increase TA and decrease upon oxidation (or biological uptake), if no biological effect is involved, there is no TA implication, i.e., deposition followed by oxidation or uptake.

Response: Good point. This is how it is implemented in the code. The text has now been adjusted to clarify this (Section 4.6).

P14, the title of Section 4.5, this is not implications, rather simulations of future scenarios.

Response: We agree. The title has been changed to better suit the contents.

P15 L29, please update the link.

Response: Updated, and also the reference itself has been changed and now reads: van Heuven, S., Pierrot, D., Rae, J.W.B., Lewis, E. and Wallace, D.W.R.: MATLAB Program Developed for CO2 System Calculations. ORNL/CDIAC-105b. Carbon Dioxide Information Analysis Center, Oak Ridge National Laboratory, U.S. Department of Energy, Oak Ridge, Tennessee. doi: 10.3334/CDIAC/otg.CO2SYS_MATLAB_v1.1, 2011 to van Heuven et al. (2011).

P15 L33-34, what about including possible temperature increase? It would be a better approach to simulate future pH with invariant TA, like in most studies.

Response: The temperature increase (and also the salinity decrease) is already included in the BALTSEM simulation (approximately 3 degrees C, see Section 4.5 and Fig. S5). We did not however examine the pH sensitivity to temperature change – this is an interesting topic and the BALTSEM model would indeed be a useful tool. But, this study is primarily focused on TA dynamics; we did not want to weigh it down with further sensitivity analyses. Using invariant TA has as you say been done in prior

studies, and one main topic of this study is the effect of (long-term) changes in TA. For those reasons we did not consider scenarios with invariant TA.

Please also note the supplement to this comment:
https://www.biogeosciences-discuss.net/bg-2018-313/bg-2018-313-AC3-supplement.pdf

———————————————————

[Figure]

[Figure]

**Fig. 1.** New Figure 5

[Figure]

[Figure]

[Figure]

**Fig. 2.** New Figure 6

**Supplement:**

**Primary and secondary redox reactions**

Here we present redox reactions represented by the RTM (Table S1-S2) and BALTSEM (Table S3), respectively, that are important in terms of TA dynamics and buffering capacity. A more complete description of sediment reactions is available in the supplementary material by Reed et al. (2016). Organic material (OM) in the reactions below is defined as $(CH_2O)_a(NH_3)_b(H_3PO_4)_c$, where a:b:c defines the molar C:N:P ratios for OM. $\lambda \approx 0.18$ is the P:Fe ratio for iron-bound phosphorus (Reed et al., 2016).

TA in the RTM is defined according to (see Section 2.2.1 of corresponding article):

$$TA = [HCO_3^-] + 2[CO_3^=] + [HPO_4^=] + 2[PO_4^{3-}] + [NH_3] + [HS^-] - [H^+]$$

TA in BALTSEM is defined according to (see Section 2.2.2 of corresponding article):

$$TA = [HCO_3^-] + 2[CO_3^=] + [B(OH)_4^-] + [HPO_4^=] + 2[PO_4^{3-}] + [SiO(OH)_3^-] + [NH_3] + [HS^-] - [H^+] - [HF] - [H_3PO_4] + organic\ alkalinity$$

**Tables and figures**

**Table S1. Primary redox reactions implemented in the RTM and the related changes in TA ($\Delta$TA).**

| Process | $\Delta$TA |
|---|---|
| Aerobic mineralization $$OM + aO_2 \longrightarrow aCO_2 + aH_2O + bNH_3 + cH_3PO_4$$ | b – c |
| Denitrification $$OM + \frac{4a}{5}NO_3^- \longrightarrow \frac{a}{5}CO_2 + \frac{4a}{5}HCO_3^- + \frac{3a}{5}H_2O + \frac{2a}{5}N_2 + bNH_3 + cH_3PO_4$$ | 0.8a + b – c |
| Manganese oxide (MnO$_2$) reduction $$OM + 2aMnO_2 + 3aCO_2 + aH_2O \longrightarrow 4aHCO_3^- + 2aMn^{2+} + bNH_3 + cH_3PO_4$$ | 4a + b – c |
| Iron oxyhydroxide (Fe(OH)$_3$) reduction $$OM + 4a\{Fe(OH)_3 - \lambda HPO_4^=\} + 7aCO_2 \longrightarrow 8aHCO_3^- + 3aH_2O + 4aFe^{2+} + bNH_3 + cH_3PO_4 + 4a\lambda HPO_4^=$$ | 8a + b – c + 4a$\lambda$ |
| Sulfate (SO$_4$$^=$) reduction $$OM + \frac{a}{2}SO_4^= \longrightarrow aHCO_3^- + \frac{a}{2}H_2S + bNH_3 + cH_3PO_4$$ | a + b – c |
| Methanogenesis $$OM \longrightarrow \frac{a}{2}CO_2 + \frac{a}{2}CH_4 + bNH_3 + cH_3PO_4$$ | b – c |

**Table S2. Secondary redox reactions implemented in the RTM and the related changes in TA (ΔTA).**

| Process | ΔTA |
|---|---|
| Nitrification
$2O_2 + NH_4^+ + 2HCO_3^- \longrightarrow NO_3^- + 2CO_2 + 3H_2O$ | -2 |
| Manganese re-oxidation
$O_2 + 2Mn^{2+} + 4HCO_3^- \longrightarrow 2MnO_2 + 4CO_2 + 2H_2O$ | -4 |
| Iron re-oxidation
$O_2 + 4Fe^{2+} + 8HCO_3^- + 2H_2O + 4\lambda HPO_4^= \longrightarrow 4\{Fe(OH)_3 - \lambda HPO_4^=\} + 8CO_2$ | $-8 - 4\lambda$ |
| Iron monosulfide (FeS) re-oxidation
$2O_2 + FeS \longrightarrow SO_4^= + Fe^{2+}$ | 0 |
| Pyrite re-oxidation
$7O_2 + FeS_2 + 2H_2O \longrightarrow 4SO_4^= + 2Fe^{2+} + 4H^+$ | -4 |
| Dissolved sulfide (H$_2$S) re-oxidation
$2O_2 + H_2S + 2HCO_3^- \longrightarrow SO_4^= + 2CO_2 + 2H_2O$ | -2 |
| Aerobic methane oxidation
$2O_2 + CH_4 \longrightarrow CO_2 + 2H_2O$ | 0 |
| Iron re-oxidation coupled to manganese oxide reduction
$MnO_2 + 2Fe^{2+} + 2\lambda HPO_4^= + 2H_2O + 2HCO_3^- \longrightarrow 2\{Fe(OH)_3 - \lambda HPO_4^=\} + Mn^{2+} + 2CO_2$ | $-2 - 2\lambda$ |
| Manganese oxide reduction using dissolved sulfide
$MnO_2 + H_2S + 2CO_2 \longrightarrow Mn^{2+} + S_0 + 2HCO_3^-$ | 2 |
| Iron oxyhydroxide reduction using dissolved sulfide
$2\{Fe(OH)_3 - \lambda HPO_4^=\} + H_2S + 4CO_2 \longrightarrow 2Fe^{2+} + 2\lambda HPO_4^= + S_0 + 4HCO_3^- + 2H_2O$ | $4 + 2\lambda$ |
| Iron monosulfide (FeS) formation
$Fe^{2+} + H_2S + 2HCO_3^- \longrightarrow FeS + 2H_2O + 2CO_2$ | -2 |
| Anaerobic CH$_4$ oxidation by SO$_4^=$ reduction
$SO_4^= + CH_4 + CO_2 \longrightarrow 2HCO_3^- + H_2S$ | 2 |
| Elemental sulfur disproportionation
$4S_0 + 4H_2O \longrightarrow 3H_2S + SO_4^= + 2H^+$ | -2 |
| Pyrite (FeS$_2$) formation using dissolved sulfide
$FeS + H_2S \longrightarrow FeS_2 + H_2$ | 0 |
| Pyrite (FeS$_2$) formation using elemental sulfur
$FeS + S_0 \longrightarrow FeS_2$ | 0 |
| Vivianite (Fe$_3$(PO$_4$)$_2$·8H$_2$O) formation
$3Fe^{2+} + 2HPO_4^= + 8H_2O \longrightarrow Fe_3(PO_4)_2 \cdot 8H_2O + 2H^+$ | -4 |
| Transformation from amorphous (α) to well-crystalline (β) iron oxyhydroxide (Fe(OH)$_3$)
$Fe(OH)_3^\alpha \longrightarrow Fe(OH)_3^\beta$ | 0 |
| Transformation from amorphous (α) to well-crystalline (β) manganese oxide (MnO$_2$)
$MnO_2^\alpha \longrightarrow MnO_2^\beta$ | 0 |

**Table S3. Reactions included in BALTSEM and the related changes in TA (ΔTA).**

| Process | ΔTA |
|---|---|
| Primary production | |
| $aCO_2 + (a + b)H_2O + bHNO_3 + cH_3PO_4 \rightarrow OM + (a + 2b)O_2$ | b + c |
| $aCO_2 + aH_2O + bNH_3 + cH_3PO_4 \rightarrow OM + aO_2$ | -b + c |
| | |
| Aerobic mineralization | |
| $OM + aO_2 \rightarrow aCO_2 + aH_2O + bNH_3 + cH_3PO_4$ | b – c |
| | |
| Denitrification | |
| $OM + \frac{4a}{5}NO_3^- \rightarrow \frac{a}{5}CO_2 + \frac{4a}{5}HCO_3^- + \frac{3a}{5}H_2O + \frac{2a}{5}N_2 + bNH_3 + cH_3PO_4$ | 0.8a + b – c |
| | |
| Sulfate ($SO_4^=$) reduction | |
| $OM + \frac{a}{2}SO_4^= \rightarrow aHCO_3^- + \frac{a}{2}H_2S + bNH_3 + cH_3PO_4$ | a + b – c |
| | |
| Nitrification | |
| $2O_2 + NH_4^+ + 2HCO_3^- \rightarrow NO_3^- + 2CO_2 + 3H_2O$ | -2 |
| | |
| Dissolved sulfide ($H_2S$) re-oxidation | |
| $2O_2 + H_2S + 2HCO_3^- \rightarrow SO_4^= + 2CO_2 + 2H_2O$ | -2 |

**Table S4. Equilibrium constants and associated equations for pH-dependent species.**

[revised manuscript text omitted]

---

## Author Response (AR1)

**Authors' response**

*A 'track changes'-version of the manuscript is included after the point-by-point responses.*

**Response to referee #1**

*Page and line numbers in the responses below refer to the 'track changes'-version of the manuscript.*

This manuscript reports alkalinity development in the Baltic Sea since 1970 based on two models. I am, however, not convinced about the results as there is a lot of extrapolation, and the calculations are done without an error analysis. For instance, it is said that 260 Gmol/y of the additional TA source can not be explained but there is no error bar. In fact, most of the numbers given have no error bars.

*Response: The standard deviation of all terms in the TA budget has now been added in Table 5 and also in the text (Section 3.2, P11 L13-14).*

My other major concerns are: 1. Sulfur in umol/g is converted to umol/cm3 using measured porosity but this can not be done using porosity alone.

*Response: Besides porosity, a sediment density of 2.65 g cm$^{-3}$ is used for this conversion, which is typical for marine sediments. This is now added in the manuscript (Section 2.1.1, P4 L6).*

2. The results are integrated to 25cm and extrapolated to basin scale. But, how reliable is the age model and how homogeneous is the sediments in the Baltic Sea? I would expect a lot of variability in both but the uncertainty is not evaluated.

*Response: The age model for site F80 is extensively discussed by Lenz et al. (2015b). Briefly, it was constructed using high-resolution Mo and Mn data obtained by laser ablation-inductively coupled plasma mass spectrometry (LA-ICP-MS) line scanning. By comparing fluctuations in Mo/Al and Mn/Al with instrumental records of bottom water oxygen conditions, ages were assigned to features in these profiles. It is the best age model available for this site, and as already shown by Reed et al. (2016), matches well with likely scenarios regarding bottom-water oxygenation and organic matter input. Assuming that the upper 25 cm of the sediment represents the period 1970-2009, is thus validated, in our opinion.*

*As also responded to Reviewer 3, we would like to stress that we extrapolate our results not over the entire Baltic Sea, but only over muddy sediments in the Baltic Proper (or central Baltic Sea) which is where F80 is located. However, there are indeed spatial differences in the sediment geochemistry of muddy Baltic Proper sediments. To clarify the uncertainties and limitations with our approach, the following paragraph has been added to Section 4.1 (P13 L1-8):*

*"The RTM fluxes are upscaled under the assumption that the fluxes computed for the F80 site are representative for the muddy sediment area of the Baltic Proper. This assumption is associated with uncertainties because of spatial differences in the sediment geochemistry of muddy Baltic Proper sediments as illustrated by the pore water and Fe-S chemistry for 4 other sites as published by Lenz et al. (2015). The solid phase profiles for these sites show similar*

*temporal trends over the past decades as F80. Furthermore, the pore water profiles show that site F80 has a relatively high rate of organic matter deposition and alkalinity regeneration when compared to most of the other sites. This implies that, with our extrapolation, the role of the sediment could be slightly overestimated.  Thus, the large-scale fluxes we obtain by extrapolating fluxes from one specific site are to be regarded as a maximum estimate of the contribution of S burial to the overall TA budget of up to 26%".*

3. It is said that borate is ignored because it is expected to have a low contribution to TA. I am surprised. What is the basis for such a statement?

*Response: Borate is included in BALTSEM as indicated in the new Equation 4 (P6 L19-20) as well as in the supplementary material; the contribution in most areas is small because of the low salinity, although one exception is the entrance area with more oceanic conditions. In the RTM on the other hand, borate is not included as its contribution to TA is even smaller. With S=12 (i.e. total boron = 142.5 umol/kg according to Uppström, 1974) and prevailing conditions, bottom-water borate alkalinity (BA) is ~9e-3 mM, while measured TA equals 2.2 mM. Thus, BA contributes only 0.4% to TA in the bottom-water. With no significant source or sink term for boron in the sediments, its contribution to TA sharply declines with sediment depth. It is thus acceptable to ignore this term in the RTM.*

My minor concern is the statement that the Baltic Sea today forms the largest anthropogenic dead zone is the world. It is not true. There are many dead zones in the Baltic Sea so the authors must have summed them up. If one sums up separate dead zones in other seas, such as the East China Sea, the total area is much larger. In fact, the dead zone off the Changjiang river mouth alone is now larger.

*Response: We now write "one of the largest" instead of "the largest" (Section 1, P2 L33).*

**Response to referee #2**

*Page and line numbers in the responses below refer to the 'track changes'-version of the manuscript.*

Overall Statements

The manuscript "Sedimentary alkalinity generation and long-term alkalinity development in the Baltic Sea" by Erik Gustafsson and colleagues presents the simulated development of alkalinity generation in the Baltic Sea over the last decades and, additionally, projections until 2100. The modelling tools include a reactive-transport model (RTM) for sedimentary processes which is able to resolve Fe-S cycling and burial of corresponding components, which in turn generates TA. Such irreversible processes are necessary to describe the missing (unresolved) contributors to the overall TA sources in the Baltic Sea.

Instead of a coupled physical – biogeochemical 3D model which couples benthic and pelagic processes, the authors use the less expensive model BALTSEM for the different Baltic Sea basins and the RTM which is weakly coupled to BALTSEM. For the reader it is unclear which information (fluxes) are provided for RTM by BALTSEM and vice versa. A full bidirectional coupling of both models, which is claimed as not feasible (I doubt) will definitively produce results differing from the weak applied coupling. It is necessary to estimate the error induced by this weak coupling. I suggest to test this with an application of BALTSEM for one water column and the underlying RTM sediment core. Within one scenario the weak coupling should be applied and within another scenario a full coupling should run. With these two results the authors can compare the TA generation of both scenarios and hopefully are able to demonstrate that the result of the weak coupling shows the main TA-related features as the full coupled run.

*Response: One main problem with the implementation of a full coupling between the two models is that the same state variables would have to be included in both models. This would in particular require a completely new version of BALTSEM that includes e.g. $Fe^{2+}$, $Fe(OH)_3$, $Mn^{2+}$, $MnO_2$, $S_0$, $FeS$, $FeS_2$, etc. For each new state variable, we would furthermore need external loads and boundary conditions. This is not impossible but a massive undertaking and not a realistic goal for the time being. Since the external loads are poorly known, this would in addition add large uncertainties. Developing a BALTSEM version with just one water column would not remove such obstacles.*

*To clarify these issues, the last paragraph of Section 2.2.3 (P8 L19-27) now reads: "Ideally the RTM would be dynamically coupled to BALTSEM, but this is currently not feasible for two reasons: First and foremost, direct coupling would require that the state variables used in the two models would have to match so that the same reactions can be simulated in both models. This means that we would have to add numerous new state variables to BALTSEM (cf. Table S1-S3). For each new state variable BALTSEM would furthermore need external loads and boundary conditions. Implementation of a full coupling between the two models is in other words a massive task and far beyond the scope of this study. Second, BALTSEM has approximately 1400 sediment "boxes", and the RTM would have to compute the sediment processes in each of these boxes – calibration of the RTM in various parts of the Baltic Sea would be problematic*

*because of an insufficient coverage of sediment data. Therefore, the two models are not directly coupled to one another but instead used independently".*

*A realistic future goal would be to develop an intermediate version of BALTSEM that includes the aspects of sedimentary Fe-S cycling that we believe to be crucial, but not the detail that is possible in the RTM. Section 4.1 (P13 L9-14) has been updated with the following paragraph: "Although a full coupling between the two models is not a realistic goal at the moment, the development of sediment processes in BALTSEM is decidedly a highly desirable future goal. In particular, the inclusion of sedimentary Fe-S dynamics and related phosphorus (P) cycling would serve to improve our understanding of both TA and P dynamics on a system scale. The present study can be seen as an intermediate step towards a more detailed (if not complete) model description of sediment processes in the Baltic Sea. In fact, the relatively large influence of sedimentary processes on TA dynamics that we demonstrate in this study also serves as a motivation to pursue this goal".*

*The two models are used independently which means there are no new errors induced by a weak coupling. The two models do of course have individual shortcomings and uncertainties.*

*The RTM is used to estimate how much the sedimentary Fe-S cycling could contribute to the Baltic Sea TA budget. Results are then compared to the large-scale BALTSEM model and this exercise quantitatively demonstrates the importance of sedimentary processes compared to other (external and internal) TA sources and sinks. To clarify what processes are included in which model, we have added three new tables (Table S1- S3) where the processes and state variables included in each model are listed (this is also clearly indicated in the model descriptions in Section 2.2.1 and 2.2.2 respectively).*

One of the main conclusions of the manuscript is that Fe-S dynamics impact the TA generation only on longer time scales. This is derived from one sentence on page 15 line 23. For this conclusion I expect a deeper analysis.

*Response: We do not believe that this is one of the main conclusions of the manuscript, and we are not sure how the reviewer arrived at this interpretation of the manuscript. P18 L22-24 says: "Lowering the Fe-oxide loading to pre-1973 values decreases the S burial by an order of magnitude, confirming its limitation by Fe. Strikingly, the TA efflux is only marginally impacted, indicating again the decoupling between short-term flux dynamics and long-term TA generation, as discussed extensively in Sections 2.2.3 and 3.1" What we discuss here is the mismatch between sedimentary TA generation and the modelled effluxes of TA.*

*Instead, one of the main conclusions is that burial of Fe sulfides is a major process impacting long-term TA generation in the Baltic Sea. This indicates both different spatial and temporal scales than the reviewer's statement; one should interpret long-term TA generation as the net TA generation, i.e. the TA change occurring after all re-oxidation reactions took place, in the coupled water column-sediment system. TA generation through various processes at a specific moment in time within different zones in the sediment is highly impacted by Fe-S dynamics, as e.g. Table 4 and Figure 3 show, and as is discussed extensively in Section 3.1.*

*We have sharpened our language in the revised manuscript in such a way that this confusion cannot arise.*

Detailed remarks

P2 L3: Sarmiento and Gruber, 2006: Ref missing

*Response: Corrected.*

P2 L8: Rabalais et al., 2015: Ref says 2014

*Response: Corrected.*

P2 L17 and L25 Reference List shows only Hu and Cai, 2011

*Response: No, both are there! (This is much easier to see now that indentations have been added).*

P3 L1: Table 1 in Gustafsson et al 2014b gives 453 Gmol yr-1 as riverine TA load.

*Response: We refer to the value used in the budget calculations (Table 3 by Gustafsson et al., 2014b). The text has been slightly adjusted to clarify this.*

P4 L21: The expression ΣH2S must be introduced.

*Response: The definition ($\Sigma H_2S = [HS^-] + [H_2S]$) is now included in Section 1, P3 L8.*

P5 L2: How large was the increase of TA loads when the new Swedish and Finnish data were included?

*Response: As can be seen in Fig. S2 (supplementary material), the TA loads from Finnish rivers appear to be rather constant. However, we only have 10-year record for all Finnish rivers, so of course it is not easy to determine a trend. But, we also looked into 3 northern Finnish rivers that have 40-year data and found a generally increasing DIC flux to the Baltic. This concludes increasing weathering fluxes by 10-20% over the last 40 years (Sun et al., 2017, Chemical Geology). In Swedish rivers there is on average an increase of more than 5 Gmol over a 25-year period (Fig. S2). This is a significant increase compared to the total load from Swedish rivers ($\sim$40 Gmol yr$^{-1}$), but compared to the TA pools in the Baltic Sea (in total $\sim$33000 Gmol; Gustafsson et al., 2014b) this is of marginal importance.*

*Reference: Sun, X., Mörth, C.-M., Humborg, C. and Gustafsson, B.: Temporal and spatial variations of rock weathering and CO2 consumption in the Baltic Sea catchment, Chemical Geology, 466, 57–69, doi:10.1016/j.chemgeo.2017.04.028, 2017.*

P5 L17: Lukawska-Matuszewska and Kielczewska, 2016

*Response: The list of references accidentally mentioned an incorrect paper. It should have said: Łukawska-Matuszewska, K.: Contribution of non-carbonate inorganic and organic alkalinity to total measured alkalinity in pore waters in marine sediments (Gulf of Gdansk, S-E Baltic Sea),*

*Marine Chemistry, 186, 211–220, doi:10.1016/j.marchem.2016.10.002, 2016. This is now corrected.*

P6 L18: The use of these unresolved fluxes is very unsatisfying. They might also represent sinks that are assumed too high. Using such a "joker", it's relatively easy to match observed TA concentrations.

*Response: Yes, the unresolved fluxes could include overestimated sinks (probably a minor part though). The purpose of the simulated unresolved fluxes was to match observed TA concentrations as closely as possible and further to close the TA budget of the Baltic Sea.*

P7 L3ff: How do you handle the lateral Fe input? How do you treat S burial and the consecutive TA flux into the pelagic? The normal way across pore water diffusion in connection with overlying water cannot work with this model setup.

*Response: Variations in the lateral input of Fe have been described as variations in the amount and form of Fe deposited onto the sediment, as shown in Figure 2. The paper by Reed et al. (2016) provides more details on the choices of this; as the manuscript is already quite long as is, and the model calibration did not make up part of this work, we did not want to repeat too many of the details. Instead, we now included additional references to either Figure 2 or the work of Reed et al. (2016) in the manuscript where necessary.*

*The point the reviewer makes here is one of the reasons why we do not directly link calculated effluxes to changes in the water column. Instead, the amount of S burial in a specific year is assumed to represent a release of TA from the sediments within that year. Given the relatively long time scale that we are looking at (averages over multiple years) compared to the actual rate of formation, we can assume that all TA associated with S burial will have diffused upwards and escaped the sediment. This is discussed in Sections 2.2.3 and 3.1.*

P7 L10 Describe the upscaling process in more detail.

*Response: After translating S burial to an efflux of TA (see response to previous comment), with units of mmol TA $m^{-2}$ $y^{-1}$, we assumed this flux to be representative for the entire muddy area of the Baltic Proper. This is further detailed in Section 4.2, but it is now also specified in Section 2.2.3 (P7 L30 -P8 L10) where the first paragraph is extended as follows: "The RTM on the other hand resolves these processes in detail and quantifies the fluxes at specific sites. It is not feasible to upscale such site-specific fluxes to the system-scale. Moreover, it would require that the fate of all components contributing to the TA efflux calculated by the RTM should be evaluated in BALTSEM. We know that a substantial part of the TA efflux from the sediment is due to components that are reoxidized in the water column. Only a full coupling between both models, which is currently not feasible as discussed below, would allow monitoring the fate of these components. We therefore use only that part of the TA efflux that is due to a sedimentary source that is permanent on the time scale of interest, i.e. the burial of reduced S. In the present study, the amount of S burial in a specific year is assumed to represent a release of TA from the sediments within that year. Given the relatively long time scale that we are looking at (averages over multiple years) compared to the actual rate of formation, we can assume that all TA*

*associated with S burial will have diffused upwards and escaped the sediment. This TA flux due to S burial and computed by the RTM was subsequently upscaled to cover a certain bottom type in the relevant sub-basin (i.e., the total muddy sediment area). This was done by multiplying the net TA generation resulting from S burial (mmol m$^{-2}$ y$^{-1}$) by the muddy sediment area of the Baltic Proper (Table 1)".*

P7 L19 I do not see the problem to handle 1400 sediment "boxes".

*Response: In Section 2.2.3 (P8 L20-27) we have now clarified the main bottlenecks related to a full coupling of the models. Handling 1400 sediment boxes is one of those, but certainly not the most important one. See reply above.*

P7 L22 You should say that the current model setup is only an intermediate step towards full coupling.

*Response: A full coupling is not a realistic goal for the time being for reasons described in Section 2.2.3 (see comment above). However, an improved description of sediment dynamics in BALTSEM is a highly desirable future goal. A new paragraph discussing future goals has been added to Section 4.1 (see comment above).*

P10 L14-20 The text is non-transparent. Enumerate all shortcuts and discuss their implications. Specify the processes and species, which cannot be linked. Here, the above mentioned sensitivity study should be discussed.

*Response: The text in Section 4.1 (P12 L7-15) has now been updated with references to our new Tables S1-S3 as well as to the updated Section 2.2.3 (described in comment above):*

*"BALTSEM includes many biogeochemical processes that produce and consume TA both reversibly and irreversibly on short time scales and in many boxes within each sub-basin of the Baltic Sea. These processes are described in Section 2.2.2 and are further listed in detail in Table S3. BALTSEM furthermore accounts for land loads, atmospheric depositions, and TA exchange between sub-basins and between the Baltic Sea and the North Sea. The result of the model simulations, i.e. the long-term development of TA in various sub-basins, is what we compare to observations in the water column (Fig. 5-6). Similarly, the RTM calculates net TA generation due to various reversible and irreversible processes (described in detail in Table S1-S2). If we dynamically coupled the RTM to BALTSEM, we would have to consider all these processes, and link all species between both models. Given the unfeasibility of this, as discussed in Section 2.2.3, we couple both models by using the output of the RTM to further constrain BALTSEM. Specifically, we explain part of the source of BALTSEM that is unresolved but necessary to describe the long-term TA development in the Baltic Sea".*

P12 L24 Dijkstra et al,. 2018: Ref says 2017

*Response: The reference has been updated and now reads: Dijkstra, N., Hagens, M., Egger, M. and Slomp, C. P.: Post-depositional formation of vivianite-type minerals alters sediment phosphorus records, Biogeosciences, 15(3), 861–883, doi:https://doi.org/10.5194/bg-15-861-2018, 2018.*

P15 L23: "Striking .." Discuss this item in more detail. Why would you have assumed a stronger impact? Which mechanism hampers it?

*Response: We agree this is an interesting finding that warrants a further explanation. If we had assumed a tight coupling between S burial and modelled TA effluxes, we would have expected a decline in TA efflux as well. However, averaged over the 41-year period, we do not observe this at all, while we expected to see at least some response in the modelled TA effluxes. In part, this can be because in this scenario the amount of TA generation due to OM degradation (2855 mmol $m^{-2} y^{-1}$) exceeds by far the amount of TA generation due to S burial (46 mmol $m^{-2} y^{-1}$), also under business-as-usual (470 mmol $m^{-2} y^{-1}$). Also, because an important source of reduced S comes from below, due to $SO_4$-AOM at depth, the formation of S solids generally occurs deeper in the sediment than OM degradation. In contrast, we did see a strong decline in modelled TA effluxes when the OM loading is lowered. This suggests that the modelled TA effluxes are indeed dominated by the amount of OM degradation. On P9 L33ff we already discussed the interesting link between modelled TA efflux and changes in iron loading, and how the depth at which reactions occur is also relevant for the modelled TA efflux.*

*We already discussed earlier in the manuscript that we cannot directly use modelled TA effluxes to study the effect on long-term TA development (P9 L14ff). The modelled TA efflux is a combined effect of many TA generating (and consuming) processes occurring in the sediment. As such, it 'blurs' the signal of S burial. Most of these sedimentary TA generating (and consuming) processes are compensated for in the water column on the time scale that we are interested in, such that only the burial of S remains as the relevant process on the long term.*

P19 L6: "2014a"

*Response: Corrected.*

P19 L9: "2014b"

*Response: Corrected.*

P36 L3: Ruppin (1909): Ref missing

*Response: Corrected.*

**Response to referee #3**

*Page and line numbers in the responses below refer to the 'track changes'-version of the manuscript.*

Gustafsson et al. investigated benthic TA generation under low oxygen conditions using a coupled physical-biogeochemical model BALTSEM and a reactive-transport sediment model (RTM). They discovered that Fe sulfide burial accounts for roughly a quarter of "missing" TA that cannot be accounted for using the BALTSEM model, with the rest of the missing TA potentially coming from groundwater and river input. In addition, the authors found that Fe limitation restricts the magnitude of the Fe-S burial produced TA. Using scenario modeling, this anaerobic TA source will become even smaller because of improvement of oxygen conditions and reduced Fe input from land, hence burial of reduced sulfur. As a result, the Baltic Sea will be more prone to the acidification risks.

The major concern I have is how representative the RTM model result is across the entire Baltic Sea. The authors stated that this model calculation is very expensive if multiple stations are included. It appears that they tried to extrapolate only one station (F80) to the entire Baltic basin where this is "muddy" sediments. It is known that sediments are heterogeneous and the Baltic definitely should not be an exception. Therefore, I question the validity of their data presentation and interpretation based on the RTM model.

*Response: The calculated fluxes at F80 were not extrapolated to the muddy areas of the entire Baltic Sea. Instead, we use it to represent muddy areas in the Baltic Proper (or central Baltic Sea) which is where F80 is located. However, there are indeed spatial differences in the sediment geochemistry of muddy Baltic Proper sediments. To clarify the uncertainties and limitations with our approach, the following paragraph has been added to Section 4.1 (P13 L1-8):*

*"The RTM fluxes are upscaled under the assumption that the fluxes computed for the F80 site are representative for the muddy sediment area of the Baltic Proper. This assumption is associated with uncertainties because of spatial differences in the sediment geochemistry of muddy Baltic Proper sediments as illustrated by the pore water and Fe-S chemistry for 4 other sites as published by Lenz et al. (2015). The solid phase profiles for these sites show similar temporal trends over the past decades as F80. Furthermore, the pore water profiles show that site F80 has a relatively high rate of organic matter deposition and alkalinity regeneration when compared to most of the other sites. This implies that, with our extrapolation, the role of the sediment could be slightly overestimated. Thus, the large-scale fluxes we obtain by extrapolating fluxes from one specific site are to be regarded as a maximum estimate of the contribution of S burial to the overall TA budget of up to 26%".*

The authors also admitted that historical TA measurement in rivers may have bias due to the lack of modern analytical standard (such as CRM) and approaches. If they can add a section of error analysis and see how that could contribute to the "missing" TA in the BALTSEM model. Given that groundwater and river could both contribute large fraction of TA budget in this area and

potentially changing hydrological conditions, a lack of this analysis make the entire argument in the manuscript a bit flawed.

*Response: It is mainly the historical measurements in the Baltic Sea (and not in the rivers) that we believe are unreliable – at least in certain periods. This is partly due to methods, and (probably more important) partly due to handling of the water samples. For example, many measurements from the 1980's and early 1990's are very likely flawed (as discussed in Section 4.4).*

*The simulated river loads of TA are (as described in Section 2.1.3) based on measurements in the period 1996-2000 where we have data from most of the major rivers entering the Baltic. The reason that we use only this short period is that it is a major obstacle to achieve data from many of the large (and alkalinity-rich) continental rivers (whereas we have long time-series form Swedish and Finnish alkalinity-poor rivers). It is furthermore difficult to judge the quality of measurements in some of these rivers but we don't know how large the error might be and how much of the missing source that can be resolved if it were possible to reliably update the river loads.*

*One main purpose of this paper is to estimate how much sedimentary processes can contribute to the overall TA budget of the Baltic Sea – and in particular to what extent these processes can explain the missing link. Trying to improve the river loads has not been a goal although this is something that we very much would like to do in future studies. But again, the bottleneck here is really to achieve (high-quality) data from all major rivers in the first place.*

*Nevertheless, to clarify a bit of the uncertainty related to the parameterization of the unresolved sources, we added a second BALTSEM simulation where the unresolved sources in the various sub-basins are added as land loads instead of sediment release. Thus, the simulations now cover the two extreme cases where the unresolved source is either explained by land loads or by sediment release. We did however not want to speculate about how these contributions might vary between the different basins (see further in comment below).*

I would also add a table on the types of reactions involved in both the BALTSEM and RTM models. The authors discussed them in multiple places but sometime they were rambling, which makes it hard to follow.

*Response: We agree. In the current version, it may not be so easy for the reader to keep track of what is included in the different models and what is not included (without consulting prior studies). We have now added three new tables to the supplementary material: In Table S1-S2 we list primary and secondary redox reactions included in the RTM, while in Table S3 we list the TA influencing reactions included in BALTSEM. This is now also clearly indicated in Section 2.2.1 and Section 2.2.2.*

Scientific questions:

P3 L2-L8, the background on TA budget is not clear. First Gustafsson et al. stated that there is an additional TA source of 344 Gmol/yr is needed to close the Baltic Sea TA budget, and 260 Gmol/yr cannot be explained, while 84 Gmol/yr was resolved with 18 Gmol/yr was attributed to net sulfate reduction. At this point, it seems that the problem has been resolved (net sulfate reduction). Then on L7, it seems that the authors separated FeS burial from net sulfate reduction, although FeS involves net sulfate reduction. Therefore, these budgetary terms need better explanation.

*Response: BALTSEM calculates the net sulfate reduction (sulfate reduction minus sulfide oxidation), but this TA source is completely reversible depending on oxygen conditions. Fe-S burial is not included in BALTSEM because we are not presently capable of modelling the coupled Fe-S cycling in the sediments (Fe is not included in the model). Thus, Fe-S burial is a TA source that we cannot explicitly take into account in BALTSEM, and this is why we need the RTM where the sedimentary processes are resolved in detail. We updated the text in Section 1, P3 L7-9:*

*"The remaining 18 Gmol $y^{-1}$ resulted from net $SO_4^{2-}$ reduction ($SO_4^{2-}$ reduction – dissolved sulfide ($\Sigma H_2S = [HS^-] + [H_2S]$) oxidation) in the water column, but this fraction could be reversed in case of oxygenation of the water column. It was hypothesized that a significant fraction of the unresolved TA source could be coupled to burial of Fe sulfides as a result of anaerobic mineralization in sediments. This would then represent a fraction of the $SO_4^{2-}$ reduction that is not readily reversed through $\Sigma H_2S$ oxidation upon re-oxygenation of the water column".*

P6 L16-30, the argument is weak. If the alkalinity contribution isn't known well (L28), how would the authors feel comfortable to run the calibration and ascertain the direction of sedimentary TA flux?

*Response: As described above, we have added a second BALTSEM simulation where the unresolved sources are instead added as land loads to the different sub-basins. The following paragraph has been added to Section 2.2.2, P7 L11-17:*

*"The processes behind the unresolved TA source are not known, but there are two candidates: external loads (e.g. river loads and submarine groundwater discharge) and internal processes (pelagic and/or benthic). In theory, the source could be associated both with processes that are not included in the model (e.g. Fe-S cycling, submarine groundwater discharge) and with processes that are included but possibly not correct (e.g. river loads, nutrient cycling). Instead of speculating about contributions from various sources in the different sub-basins, we will perform two different scenarios: one case where the unresolved source is added as additional land loads, and one case were the source is added as sediment release. The magnitudes of unresolved sources in different sub-basins are identical in the two cases".*

*The resulting TA concentrations in different sub-basins are added as additional lines in Fig. 5-6, and the following paragraph is included in Section 4.2, P13 L24-30:*

*"In the two different scenarios where the unresolved source is added as land loads (full lines in Fig. 5-6) or sediment release (dashed lines in Fig. 5-6), the simulated surface water TA concentrations are very similar (Fig. 5). Deep water concentrations on the other hand differ significantly in the Gotland Sea and the Gulf of Finland but not in the other sub-basins (Fig. 6). The reason behind the rather similar results for these two different scenarios is that land loads supplied to the different basins are rapidly distributed in the well mixed surface layer, and the well mixed surface layer constitutes a large majority of the water volume. In the deeper and more isolated parts of the system, TA concentrations are lower in the "land loads" case compared to the "sediment release" case".*

P8 L23-25, First if you refer to a multiple panel figure, please try to using letters (a, b, c...) to distinguish these panels. More importantly, if sulfide is being produced, according to reaction stoichiometry the authors presented, it would lead to TA accumulation, regardless whether this sulfide goes down or not.

*Response: In the revised manuscript we have added letters to the sub-plots and corresponding parts of the text when the figure is discussed.*

*The other point that the reviewer raises here is an excellent point. The argument refers to the fact that the model overestimates the buildup of $\sum H_2S$. Indeed, the production of sulfide leads to a generation of TA if this sulfide is not reoxidized, as is the case in the current model simulation where the sulfide is being built up in the porewater (which we argue is an artefact resulting from the chosen lower boundary condition for $\sum H_2S$). We cannot assess the fate of the sulfide when it would diffuse downwards below 32 cm depth, as we suggest that actually happens. This makes it difficult to link this $\sum H_2S$ production to a permanent, long-term TA source. However, we agree that it is not likely that the $\sum H_2S$ will be reoxidized in the time period of interest. This indicates that the net TA source to the water column due to sulfide oxidation would be higher. In the revised manuscript, we address this on P9 L23-30.*

P9 L8-11, the statement that maximum CH4-driven sulfate reduction at 8 cm is the greatest is interesting and need better explanation. If you show sulfate profile, this may look like where the sulfate-methane interface (SMZ) is and may have nothing to do with the temporal variations of CH4 production.

*Response: This is a valid point raised by the reviewer. The percentage of $CH_4$-driven sulfate reduction due to upward diffusing $CH_4$, which is on average ~95% for the period 1970-2009, does change over time (on average ~98, ~98, ~98 and ~94% for the four periods 1970-1973, 1973-1978, 1978-1981 and 1981-2009, respectively), but this change is indeed relatively minor, and upward diffusing $CH_4$ still dominates. So, we agree that the change in position of the SMTZ is more important here. The temporal evolution of porewater profiles of $SO_4$ and $CH_4$ are presented in Reed et al. (2016, their Fig. 6). From this it follows that the position of the SMTZ is shifting upwards over time, especially since 1981, when $Fe(OH)_3$ and $O_2$ declined as important terminal electron acceptors. Therefore, more $SO_4$ was consumed by OM degradation, and the SMTZ shifted upwards. We will address this in the revised manuscript by replacing the NH4 and*

*PO4 plots in Fig. 3 by the SO4 and CH4 plots, and altering the text to include the above discussion.*

P9 L12 paragraph, the authors separated different depth layers to account for the reactions that have TA implications. However, they failed to show whether TA produced in these layers make it up to TA efflux to the water column. This paragraph needs better organization and clarification.

*Response: Throughout the manuscript, we argue extensively that TA effluxes are not representative for answering the research question that we are investigating (e.g. Section 2.2.3, 3.1, 4.1). We therefore decided not to discuss the TA efflux in too much detail. We discussed that processes that impact the TA efflux are not the same processes that impact S burial or total TA generation, but that the sediment depth at which processes take place also plays a role here.*

Technical comments:

Indentation of paragraphs would be useful

*Response: Indentations are now added.*

P2 L20-L25, these sentences are a bit confusing. "On a system scale" is mentioned twice (L20 and L24). The two examples after "For examples" are essentially the same thing, so it's not clear why the authors used "however" to connect the two sentences.

*Response: This section is now rewritten, and the paragraph reads as follows (P2 L21-29):*

*"The ultimate buildup of TA due to primary production and mineralization depends on the source of the reactants and/or the fate of the products of all alkalinity-generating/consuming reactions. For example, production of dinitrogen gas ($N_2$) during pelagic or benthic denitrification results in a permanent loss of nitrate ($NO_3^-$) and hence a gain of TA (Soetaert et al., 2007). On a system scale this process only results in net TA production if the $NO_3^-$ is derived from an external source rather than from local nitrification (Hu and Cai, 2011b). Similarly, $SO_4^{2-}$ reduction leads to net TA generation only if the produced sulfide is buried as e.g. Fe sulfides rather than being reoxidized within the same system (Hu and Cai, 2011a). Note that the location of sulfide reoxidation, i.e. sediment or water column, impacts net TA generation in the sediments but not on a system scale".*

P3 L3, "unresolved" TA is 260 Gmol/yr, while in P9 L22 this number is 257.5 Gmol/yr, please be consistent.

*Response: Corrected.*

P3 L20, the two paragraphs need a better transition. Elemental sulfur seems to be introduced into the context suddenly. Prior to doing so, the authors need to explain a bit of the type of reduced sulfur burial, i.e., sulfide (with Fe) and elemental sulfur. Relative abundance of these two types of reduced sulfur also needs to be presented based on the literature.

*Response: With S we indicate (reduced) solid sulfur in any form, not only elemental sulfur. For the rest, this comment ties in with the comment below, where it is suggested to move the text of P4 L6ff to the introduction. When doing so, we directly explain why for this work it is not relevant in which form the elemental sulfur is present.*

P4 L1, is it integrated to 25 cm "interval"?

*Response: Yes, we mean the interval between 0 and 25 cm sediment depth. This is now rewritten (P4 L7).*

P4 L6-L15, this should go into the Introduction, as really this should be background information, by no means materials and methods.

*Response: We partly agree with the reviewer here, however, we are aware that various papers use various reaction equations and stoichiometries to link sulfur burial / pyrite production, mostly depending on the temporal and / or spatial scale of investigation. For example, in Łukawska-Matuszewska and Graca (2018) a different reaction equation was used for this. Therefore, we think it is important to explain the rationale behind our choice of reaction equations, and the materials and methods seems like a proper place to do so.*

P4 L24, move "for that reason" to the beginning of the sentence.

*Response: Ok.*

P4 L25, where exactly does sigma-H2S come from?

*Response: This is now explained in Section 1, P3 L8 ($\Sigma H_2S = [HS^-] + [H_2S]$).*

P5 L5 paragraph, while the authors referred to a supplemental table in a published study, it would be desirable to include such information in their own supplemental materials and refer to it in the context. Otherwise the readers can easily get lost.

*Response: Reactions included in the RTM and BALTSEM respectively are now described in Table S1-2 (RTM) and Table S3 (BALTSEM), respectively (see comment above).*

P5 L17, not a complete sentence.

*Response: We are not sure which sentence this refers to; this section seems fine.*

P7 Line 12, what is "This" approach?

*Response: This is now rewritten.*

P7 L25 and P8 L10, why are the two sulfur burial in the same time period (1970-2009) different?

*Response: Part of the sulfur solids is derived from settling of water-column particles onto the sediment surface. This cannot be derived by looking at the S solids concentrations alone (P8 L30), but becomes clear when investigating the inputs of $FeS_2$ to the sediment (Figure 2) and in-situ reaction rates (Table S6), as is done with the RTM. For the net TA generation, however, the*

*total S burial is most important, because the burial prevents possible re-oxidation in either sediment or water column and thus represents the long-term TA source that we are interested in.*

P11 L9, where is 166 Gmol/yr in Table 5?

*Response: It is there, sub-basin 7-9. The description of the division into different sub-basins and larger areas has been improved in the text. This is now described in the caption of Fig. 1 and also in the text (Section 2.1.1 and Section 4.2). The legends of Table 1 and 5 have been updated as well for clarity.*

P11 L29 and P9 L 8, I believe 43% and 43.8% in these two places are the same thing, please be consistent, at least don't round off the number in the wrong way.

*Response: This is correct and must have been a typo, as the correct (non-rounded) percentage is 43.3%. We will use ~43% on both occasions.*

P11 Section 4.2, both unit TA (mmol/m2/yr) and overall TA flux (Gmol/yr) are presented, I'd suggest that you stick with one or both (with parentheses) to avoid confusion and don't let the readers to do the conversion.

*Response: Throughout the manuscript, we stick to mmol/m2/y when discussing the sedimentary fluxes. Only when we link the results to BALTSEM, like in this paragraph, we convert them to Gmol/y.*

P12 L1 and P11 L31, this sounds like freshwater environment but not typical marine since sulfate is depleted but there is still oxidized Fe.

*Response: Iron-mediated anaerobic oxidation of methane has found to be significant in deeper Baltic Sea sediments that were subjected to a lake-marine transition (Egger et al., 2017), as well as in coastal sediments of the Bothnian Sea, which has low salinity (Rooze et al., 2016). Indeed, these conditions are not typical marine, so discussing this process is not too relevant in the context of this manuscript. We included it to be as complete as possible..*

P14 L6, not precise, it's the deposition of SOx and NOx that contribute to TA reduction, ammonia deposition first increase TA and decrease upon oxidation (or biological uptake), if no biological effect is involved, there is no TA implication, i.e., deposition followed by oxidation or uptake.

*Response: Good point. This is how it is implemented in the code. The text has now been adjusted to clarify this (Section 4.4, P17 L5-10).*

P14, the title of Section 4.5, this is not implications, rather simulations of future scenarios.

*Response: We agree. The title has been changed to better suit the contents.*

P15 L29, please update the link.

*Response: Updated, and also the reference itself has been changed and now reads: van Heuven, S., Pierrot, D., Rae, J.W.B., Lewis, E. and Wallace, D.W.R.: MATLAB Program Developed for $CO_2$ System Calculations. ORNL/CDIAC-105b. Carbon Dioxide Information Analysis Center,*

*Oak Ridge National Laboratory, U.S. Department of Energy, Oak Ridge, Tennessee. doi: 10.3334/CDIAC/otg.CO2SYS_MATLAB_v1.1, 2011 to van Heuven et al. (2011).*

P15 L33-34, what about including possible temperature increase? It would be a better approach to simulate future pH with invariant TA, like in most studies.

*Response: The temperature increase (and also the salinity decrease) is already included in the BALTSEM simulation (approximately 3 degrees C, see Section 4.5 and Fig. S5). We did not however examine the carbonate system sensitivity to temperature change – this is an interesting topic and the BALTSEM model would indeed be a useful tool. But, this study is primarily focused on TA dynamics; we did not want to weigh it down with further sensitivity analyses. Using invariant TA has as you say been done in prior studies, and one main topic of this study is the effect of (long-term) changes in TA. For those reasons we did not consider scenarios with invariant TA.*

[revised manuscript text omitted]